# C/EBPα is crucial determinant of epithelial maintenance by preventing epithelial-to-mesenchymal transition

Ana Rita Lourenço[1,2], M. Guy Roukens[1,2,4], Danielle Seinstra[3,4], Cynthia L. Frederiks[1,2,4], Cornelieke E. Pals[1,2], Stephin J. Vervoort[1,2], Andreia S. Margarido[3], Jacco van Rheenen[3] & Paul J. Coffer[1,2]*

Extracellular signals such as TGF-β can induce epithelial-to-mesenchymal transition (EMT) in cancers of epithelial origin, promoting molecular and phenotypical changes resulting in pro-metastatic characteristics. We identified C/EBPα as one of the most TGF-β-mediated downregulated transcription factors in human mammary epithelial cells. C/EBPα expression prevents TGF-β-driven EMT by inhibiting expression of known EMT factors. Depletion of C/EBPα is sufficient to induce mesenchymal-like morphology and molecular features, while cells that had undergone TGF-β-induced EMT reverted to an epithelial-like state upon C/EBPα re-expression. In vivo, mice injected with C/EBPα-expressing breast tumor organoids display a dramatic reduction of metastatic lesions. Collectively, our results show that C/EBPα is required for maintaining epithelial homeostasis by repressing the expression of key mesenchymal markers, thereby preventing EMT-mediated tumorigenesis. These data suggest that C/EBPα is a master epithelial "gatekeeper" whose expression is required to prevent unwarranted mesenchymal transition, supporting an important role for EMT in mediating breast cancer metastasis.

[1] Center for Molecular Medicine, University Medical Center Utrecht, Utrecht, The Netherlands. [2] Regenerative Medicine Center, Uppsalalaan 8, Utrecht, The Netherlands. [3] Department of Molecular Pathology, Oncode Institute, Netherlands Cancer Institute, Plesmanlaan 121, Amsterdam, The Netherlands. [4]These authors contributed equally: M. Guy Roukens, Danielle Seinstra, Cynthia L. Frederiks. *email: pcoffer@umcutrecht.nl

The epithelial-to-mesenchymal transition (EMT) is a highly dynamic and reversible process important during embryonic development, tissue repair, cancer progression, and chemotherapy resistance[1–3]. Although the pathological contribution of EMT during metastasis is currently under debate, recent studies have shown that, by a complex multistep molecular program, EMT clearly contributes to the shift of a polarized epithelial phenotype toward a mesenchymal state, where cells acquire migratory and invasive properties[4,5]. Several key signaling pathways, including TGF-β, Wnt, or Notch, have been shown to induce EMT[6]. Binding of TGFβ to its cognate receptor results in a multistep phosphorylation cascade of both receptors and key players, the Smad2 and Smad3 transcription factors (TFs), where they regulate expression of crucial genes involved in the initiation of the EMT program, including *SNAI1*, *ZEB1*, and *TWIST1*[7]. These transcription factors, known as EMT-TFs, are able to regulate cellular processes, including extracellular matrix remodeling, cell adhesion and migration, and angiogenesis[8]. The majority of these EMT-effectors are responsible for transcriptional repression of CDH1 (E-cadherin) and induction of N-cadherin expression, which contributes to weaker cell–cell interactions facilitating cell motility and invasion[9]. In addition, mesenchymal markers such as fibronectin and matrix metalloproteinases (MMPs) are upregulated during EMT. Fibronectin is important for transmitting signals from the extracellular matrix (ECM) to the cell cytoskeleton, and MMPs contribute to the remodeling of the ECM by degrading components within the ECM and cell–ECM adhesion proteins[9,10]. Hence, TGF-β signaling can contribute to EMT by inducing gene expression of transcription factors involved in epithelial integrity disruption, and acquisition of mesenchymal traits contributing to cytoskeleton reorganization, ECM remodeling, and cell motility.

The transcription factor CCAAT/enhancer-binding protein alpha (C/EBPα) is the founding member of a subfamily of the basic region leucine zipper (bZIP) transcription factors[11]. It plays a pivotal role in the regulation of cell cycle and in the expression of several lineage-specific genes[12]. Mutations in the *CEBPA* gene have been described in around 10% of acute myeloid leukemia (AML), establishing a tumor suppressor role of C/EBPα in cancer development[13,14]. In addition, extensive sequencing studies revealed that *CEBPA* is one of the 125 genes that when altered by intragenic mutations can contribute to tumorigenesis[15]. Deregulated C/EBPα expression has been observed in a variety of solid tumors such as liver, breast, or lung cancer; however, the relevance of this for tumorigenesis remains largely unclear[16].

Analysis of C/EBPα expression in healthy breast tissue and breast carcinomas has revealed that C/EBPα levels are reduced in the majority of breast cancer specimens[17]. The functional consequences of these observations remain, however, unknown. Here, we identify C/EBPα as a crucial transcription factor in maintaining epithelial architecture of human mammary cells, preventing epithelial-to-mesenchymal transition and thereby acting as a repressor of breast cancer progression in vivo.

## Results

### *CEBPA* is a SMAD3-repressed target during TGF-β-mediated EMT.
To identify novel transcription factors with a potential epithelial-gatekeeper function, we performed global RNA-sequencing analysis of TGF-β-treated or untreated human epithelial mammary (HMLE) cells. HMLE cells have been extensively used to study EMT, as they can undergo molecular and phenotypical changes upon TGF-β treatment resulting in the acquisition of mesenchymal features (Fig. 1a, Supplementary Fig. 1a)[18,19]. To specifically identify transcription factors whose expression is repressed by TGF-β, genes displaying significantly

reduced expression were analyzed by Gene-Ontology (GO)-term analysis using DAVID. Further analysis of "transcription factor activity" category revealed that E2F2, C/EBPα, and E2F8 comprise the three transcription factors that are most strongly affected by TGF-β treatment (Fig. 1b). Considering that E2F2 and E2F8 are widely expressed cell cycle regulators, we choose to further explore the role of C/EBPα as it has been shown to play a role in cell differentiation, especially during myelopoiesis, adipogenesis, and lung maturation[16,20]. Analysis of RNA-seq profiles of the *CEBPA* locus show a strong decrease in *CEBPA* mRNA levels upon TGF-β stimulation (Supplementary Fig. 1b). Furthermore, analysis of the microarray data obtained from Taube et al.[21] also showed decreased *CEBPA* expression in TGF-β1-overexpressing HMLE cells compared with control cells (Supplementary Fig. 1c). To evaluate potential redundancy between the closely related C/EBPα and C/EBPβ, we analyzed the RNA-seq profile of *CEBPB* locus in TGF-β-treated and untreated HMLE cells (Supplementary Fig. 1d). *CEBPB* expression was unaltered upon TGF-β stimulation (Supplementary Fig. 1d). Likewise, *CEBPB* mRNA levels were barely changed in HMLE cells treated with TGF-β for 24 h (Supplementary Fig. 1e). To determine the effect of TGF-β on C/EBPα expression during EMT, HMLE cells were left untreated or treated with TGF-β for 15 days, and mRNA and protein were isolated at the indicated time points (Fig. 1c, d). The EMT program was effectively induced by TGF-β as illustrated by the increase of mesenchymal markers CDH2 (N-cadherin), FN1 (Fibronectin), VIM (Vimentin), SNAI1 (Snail), and ZEB1, and decrease of epithelial marker CDH1 (E-cadherin) (Fig. 1d, Supplementary Fig. 1a). *CEBPA* mRNA expression was rapidly reduced upon TGF-β treatment, and reduced levels of *CEBPA* are maintained throughout the 15 days (Fig. 1c; Supplementary Fig. 1f). Furthermore, on the protein level, the expression of C/EBPα full-length (p42) was also found decreased upon TGF-β-mediated EMT (Fig. 1d; Supplementary Fig. 1g). Endogenous expression of C/EBPα p30 isoform in the cell lines utilized was undetectable. Unlike C/EBPα p42 isoform, shorter C/EBPα isoform lacks two critical transactivation domains (TAD1 and TAD2) responsible for interaction with elements of the RNA polymerase II basal transcriptional machinery, and therefore is unable to initiate transcription. As the decrease in C/EBPα protein levels upon TGF-β treatment was somewhat delayed compared with mRNA, we treated HMLE cells with TGF-β for shorter time points and assessed C/EBPα expression and cellular localization by immunofluorescence analysis. Immunostaining results show a gradual increase in cytoplasmic relocalization of C/EBPα after 8–72 h incubation of TGF-β (Supplementary Fig. 1h). This suggests that in addition to a rapid transcriptional downregulation of C/EBPα by TGF-β, C/EBPα can also be regulated at the posttranslational level leading to its gradual nuclear exclusion. Immunofluorescence analysis demonstrated that 15 days of TGF-β-induced EMT results in loss of C/EBPα and E-cadherin expression and increased expression levels of Fibronectin (Fig. 1e). In line with our previous observations, TGF-β treatment results in a decrease of nuclear C/EBPα levels and in an increase of C/EBPα protein localized in the cytoplasm (Fig. 1e; Supplementary Fig. 1i), suggesting that C/EBPα activity is regulated posttranscriptionally during TGF-β-mediated EMT.

Similar to HMLE cells, MCF10A cells are human non-transformed epithelial mammary cells responsive to TGF-β-stimulation[22,23]. Here, TGF-β treatment also increased the expression of mesenchymal markers including CDH2, FN1 and ZEB1, and decreased expression of the epithelial marker CDH1 (Supplementary Fig. 1j). *CEBPA* expression was also found reduced after 16 h of TGF-β treatment, and decreased expression was maintained during the 15 days (Supplementary Fig. 1j).

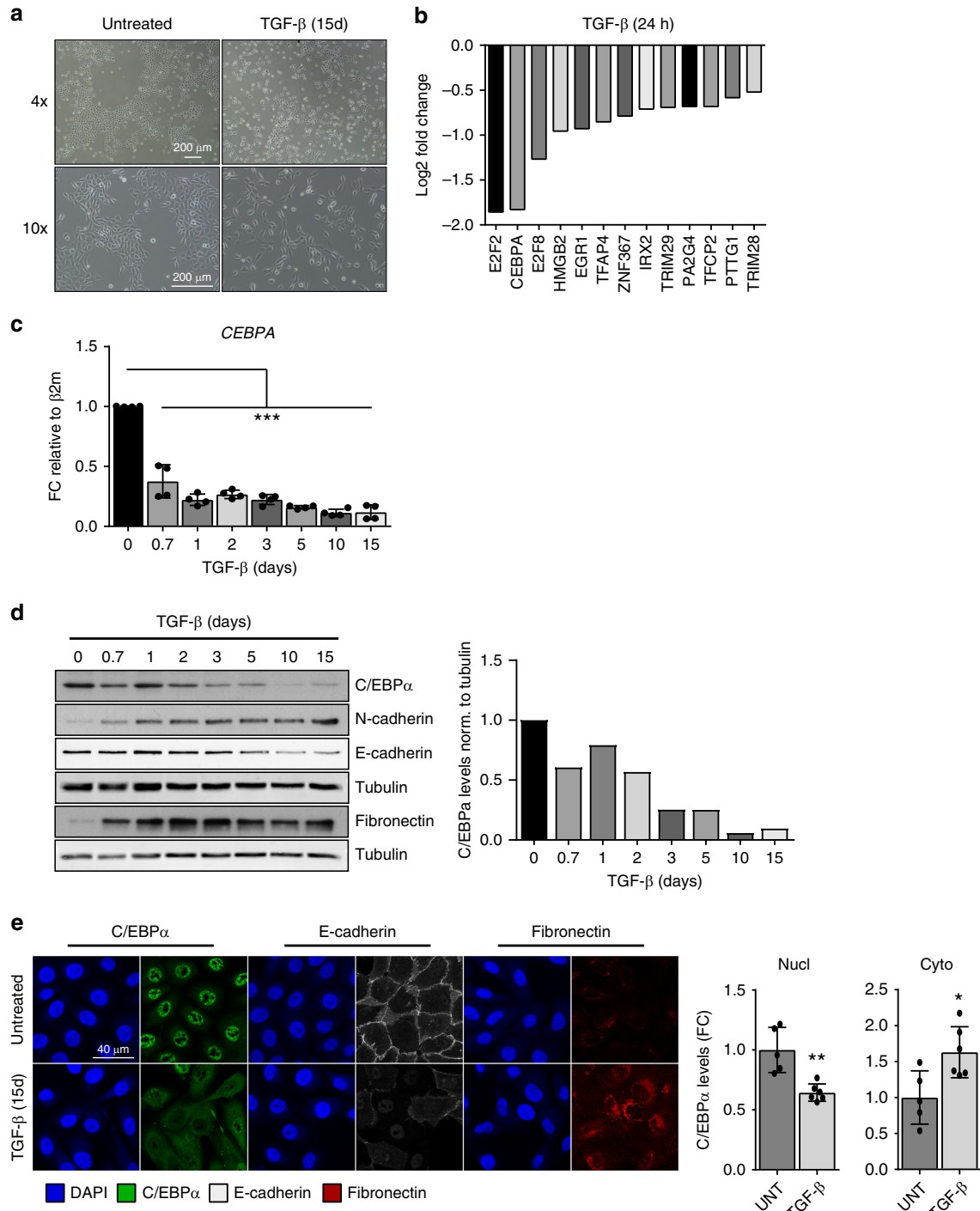

**Fig. 1 C/EBPα is a repressed target by TGF-β. a** HMLE cells were left untreated or treated with 2.5 ng/ml of TGF-β for 15 days. Bright-field microscopy images showing phenotypical changes in the presence of TGF-β signaling. Scale bar: 200 μm. Data are representative of three independent experiments. **b** HMLE cells were left untreated or treated with 2.5 ng/ml of TGF-β for 24 h. RNA samples were isolated, and purified RNA was sequenced as described in "Methods". Gene-ontology (GO) analysis were performed on the significant downregulated genes in HMLE treated cells. Transcription factor activity category and its log2 fold change is represented. **c** The results from qRT-PCR showing CEBPA mRNA expression in HMLE cells treated with 2.5 ng/ml of TGF-β as indicated. Data are represented as mean ± SD of four independent experiments. *p*-values were calculated using unpaired two-tailed Student's *t* test. ***$p < 0.001$. **d** The results from immunoblotting analysis showing the effect of TGF-β treatment on C/EBPα protein levels and established EMT markers in HMLE cells. Quantification of C/EBPα protein levels normalized to tubulin are shown. Data are representative of three independent experiments. **e** Immunofluorescence staining of C/EBPa and well-characterized EMT markers upon 15 days of TGF-β treatment in HMLE cells. Quantification of C/EBPα nuclear (nucl) and cytoplasmic (cyto) expression levels in untreated and TGF-β-treated cells is shown. Data represented as mean ± SD of five (untreated) and six (TGF-β) independent experiments. *p*-values were calculated using unpaired two-tailed Student's *t* test. *$p < 0.05$, **$p < 0.01$. d days, h hours, FC fold change, β2m beta-2-microglobulin, UNT untreated.

These data demonstrate that C/EBPα expression is rapidly and chronically downregulated by TGF-β signaling.

To determine the molecular mechanism underlying TGF-β-mediated repression of C/EBPα, we first assessed whether the binding of the TGF-β pathway effector transcription factor SMAD3 to the *CEBPA* locus was enriched upon TGF-β stimulation utilizing chromatin-immunoprecipitation sequencing (ChIP-seq). Unlike SMAD3, SMAD2 does not directly bind to DNA and therefore lacks transcriptional activity[24], providing a rationale to explore the role of SMAD3 in the repression of C/EBPα expression. ChIP-seq profile analysis of *CEBPA* locus revealed that binding of SMAD3 is increased after treatment with TGF-β (Fig. 2a). SMAD3 binding to these regions was validated using chromatin immunoprecipitation followed by qRT-PCR (ChIP-qRT-PCR). Although we observed a degree of enrichment in both regions, only SMAD3 binding on region 2 was significantly enriched in the presence of TGF-β (Fig. 2b). In order to confirm that TGF-β-mediated repression of C/EBPα is dependent on SMAD3, we generated HMLE cells expressing two independent shRNAs targeting human *SMAD3* (shSMAD3_1 and shSMAD3_2) (Fig. 2c). C/EBPα expression was effectively reduced upon TGF-β treatment in control HMLE cells (SCR), however in *SMAD3* knockdown cells, TGF-β-mediated repression of *CEBPA* was significantly impaired (Fig. 2d). Taken together, these data indicate that TGF-β-SMAD3 signaling regulates C/EBPα expression in human epithelial mammary cells and suggests that TGF-β-mediated repression of C/EBPα may be crucial for the progression of EMT.

**C/EBPα repression is crucial for TGF-β-induced EMT**. To further define the functional role of C/EBPα, we globally assessed whether constitutive C/EBPα expression could interfere with TGF-β-regulated gene expression. HMLE cells constitutively expressing C/EBPα and corresponding control cells were treated with TGF-β for 24 h, and the global transcription response was evaluated by RNA sequencing (RNA-seq). Analysis of RNA-seq data revealed 2588 differentially regulated genes after TGF-β treatment, of which 355 were also regulated by C/EBPα (Fig. 3a; Supplementary Fig. 2a). GO-term analysis of these 355 genes revealed significant enrichment for biological processes, such as cell adhesion, extracellular matrix, angiogenesis, and cell migration, all indicating a strong association with an EMT program (Fig. 3b). To confirm whether these processes were enriched in EMT, GO-term analysis was also performed using Taube et al.[21] publically available gene expression signature (known as EMT core genes). Significant enrichment of the extracellular matrix, cell adhesion, and angiogenesis was also observed by analyzing these EMT core genes (Supplementary Fig. 2b). Furthermore, to identify common transcriptional targets among the EMT core genes and C/EBPα-TGF-β core, genes were compared between both sets (Fig. 3c). This revealed a core of 37 genes included in EMT signature that can be modulated by C/EBPα expression (Fig. 3c, d). Closer examination of these 37 genes showed two main clusters: genes that are upregulated by TGF-β in control cells but repressed in the presence of C/EBPα overexpression, and genes which expression is increased in C/EBPα-overexpressing cells in both untreated and TGF-β-treated conditions. GO-Term enrichment analysis revealed that the first cluster is enriched for processes involved in cell–matrix adhesion and organization whereas second cluster is related to processes involved in inflammatory response (Supplementary Fig. 2c). We focused on genes comprising the first cluster as initial steps of EMT-induced tumor progression requires extensive cell–cell and cell–matrix reorganization[25]. To validate these targets, qRT-PCR was performed using HMLE cells expressing C/EBPα or empty vector

(EV) in the absence or presence of TGF-β. For the selected genes, suppression of TGF-β-dependent targets was observed in the presence of C/EBPα expression (Fig. 3e). In addition, vimentin (*VIM*) was utilized as a negative control since it was unaffected by C/EBPα expression (Fig. 3e).

To further evaluate the effect of C/EBPα expression during TGF-β-mediated EMT, HMLE cells overexpressing C/EBPα were treated with TGF-β for 15 days, and both mRNA and protein samples were harvested at the indicated time points. In addition, bright-field microscopy pictures were taken at the same time points. Analysis of cell morphology revealed that unlike control cells, HMLE cells overexpressing C/EBPα maintained their epithelial phenotype during TGF-β-induced EMT (Fig. 4a, arrowheads) and displayed fewer mesenchymal traits. No difference in cell growth was observed upon C/EBPα overexpression (Supplementary Fig. 3a). qRT-PCR results demonstrated that C/EBPα expression impairs TGF-β-mediated induction of mesenchymal markers including *CDH2* (N-cadherin), *MMP2*, or *FBLN5* (Fibulin-5) and repression of epithelial markers including *CDH1* (E-cadherin), *EPCAM*, and *OCLN* (occludin) supporting the phenotypical features of these cells (Fig. 4b; Supplementary Fig. 3b). Consistent with the qRT-PCR results, immunoblotting analysis showed that C/EBPα expression also reduces the TGF-β-dependent increase of mesenchymal markers such as N-cadherin and Fibronectin, and contributes to the maintenance of higher levels of E-cadherin (Fig. 4c). We further confirmed that the effect of C/EBPα on TGF-β-mediated EMT was not due to a defective TGF-β signaling pathway, since SMAD3 is equally phosphorylated upon TGF-β treatment, confirming that cells are able to signal properly upon C/EBPα overexpression (Supplementary Fig. 3c). Since C/EBPα modulates genes associated with cell migration (Fig. 3b), we asked whether C/EBPα affects the pro-migratory role of TGF-β in HMLE cells. To this end, we performed wound-healing migration assays using HMLE cells ectopically expressing C/EBPα or control empty vector in untreated or TGF-β-treated conditions. In order to mimic physiological conditions, migratory capacity of HMLE cells were evaluated on three different components of the extracellular matrix, collagen I, collagen IV, and fibronectin. In all ECM coatings, C/EBPα overexpression was sufficient to decrease wound closure of TGF-β-treated HMLE cells (Fig. 4d; Supplementary Fig. 3d). Taken together, these data strongly suggest that C/EBPα contributes to the inhibition of genes involved in epithelial-to-mesenchymal switch, and repression of C/EBPα is a requirement for initiating the EMT program.

**CEBPA knockdown impairs epithelial homeostasis**. To evaluate whether C/EBPα is a crucial determinant in maintaining epithelial integrity and thereby preventing spontaneous EMT, *CEBPA*-depleted HMLE cells were generated using two independent shRNA constructs. qRT-PCR analysis of *CEBPA* expression showed a strong reduction in *CEBPA* mRNA levels in only one of the shRNAs used (Fig. 5a), therefore HMLE cells expressing shCEBPA_1 were used for further experiments. Whereas HMLE cells expressing shRNA control (SCR) showed no phenotypical alterations, *CEBPA* knockdown cells displayed elongated shape and reduced cell–cell interactions (Fig. 5b). To assess whether morphological changes induced by *CEBPA* knockdown were associated with EMT, mRNA and protein levels of several well-established EMT markers were analyzed. qRT-PCR results revealed increased expression of mesenchymal markers, including *CDH2* and *FN1*, and decreased expression of epithelial markers in cells expressing shRNA *CEBPA* or transfected with a siRNA pool targeting *CEBPA* (Fig. 5c; Supplementary Fig. 4a). Likewise, immunoblotting analysis showed increased levels of N-cadherin and fibronectin, and reduced

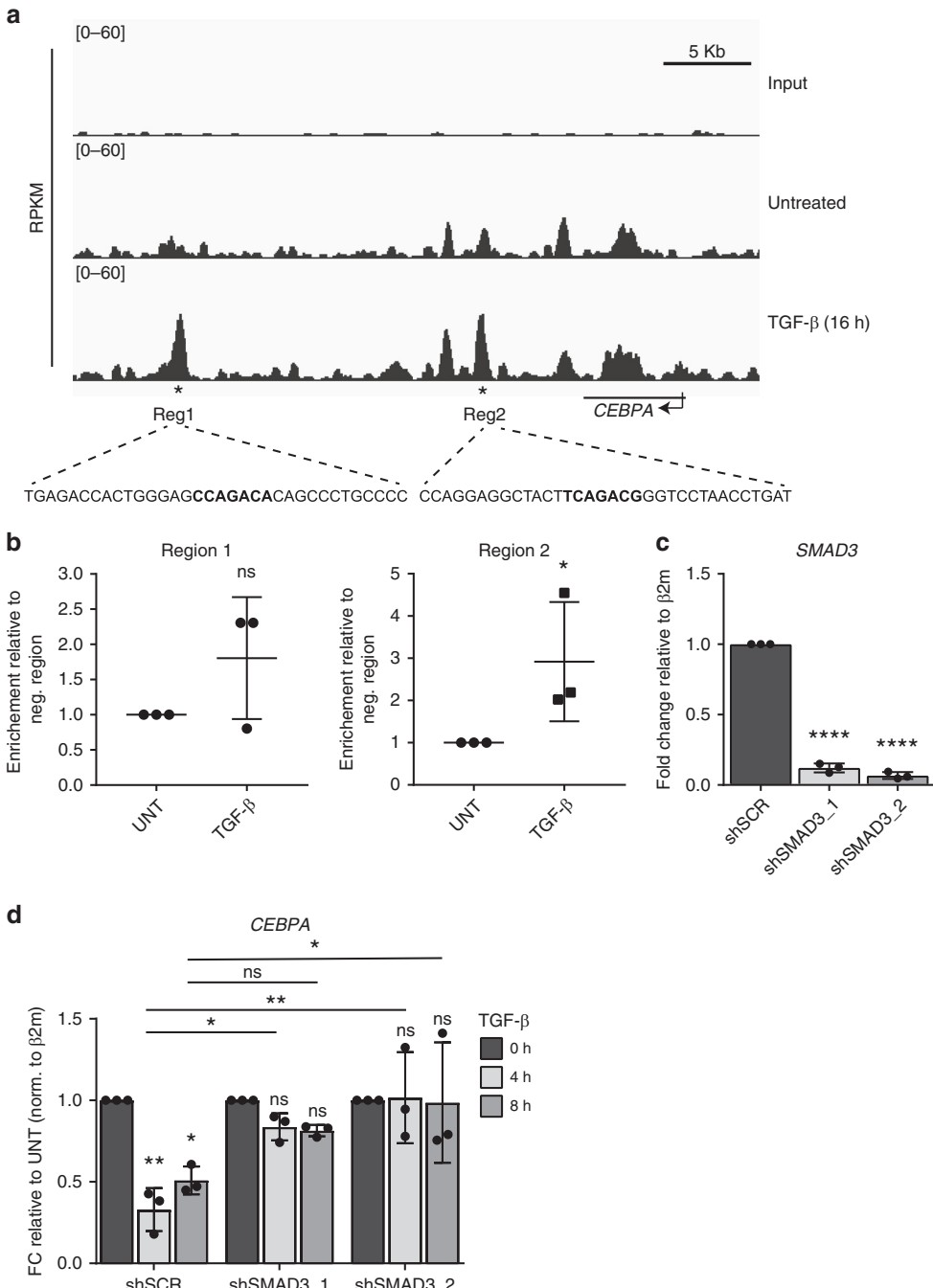

**Fig. 2 TGF-β-induced downregulation of C/EBPα is mediated by the transcription factor Smad3. a** Visualization of SMAD3 CHIP-seq profile within the genomic region surrounding the *CEBPA* locus. Stars indicate enriched picks upon 16 h of TGF-β stimulation. Part of the SMAD3-binding regions sequence is displayed, showing in bold SMAD3 known primary binding motif (C/$_T$C/$_T$A/$_C$GACA/$_G$). **b** Chromatin immunoprecipitation was performed in TGF-β treated or untreated HMLE cells using Smad3 antibody and according to "Methods". qRT-PCR was performed using specific primers targeting enriched areas (regions 1 and 2) on *CEBPA* locus. Data represented as mean ± SD of three independent experiments, normalized for a negative region. *p*-values were calculated using unpaired one-tailed Student's *t* test. *$p < 0.05$. **c** The results from qRT-PCR showing *SMAD3* mRNA levels in HMLE cells expressing shRNA control (SCR) or two independent shRNAs targeting human SMAD3 (shSMAD3_1 and shSMAD3_2). Data represented as mean ± SD of three independent experiments. *p*-values were calculated using unpaired two-tailed Student's *t* test. ****$p < 0.0001$. **d** HMLE cells expressing shSCR or shSMAD3_2 were left untreated or treated with TGF-β for 4 or 8 h. qRT-PCR analysis displaying the effect of TGF-β stimulation on *CEBPA* mRNA levels in the presence of SMAD3 knockdown. Data represented as mean ± SD of three independent experiments. *p*-values were calculated using two-way ANOVA with Tukey's multiple comparisons test. neg negatived, h hours, FC fold change, β2m beta-2-microglobulin, UNT untreated.

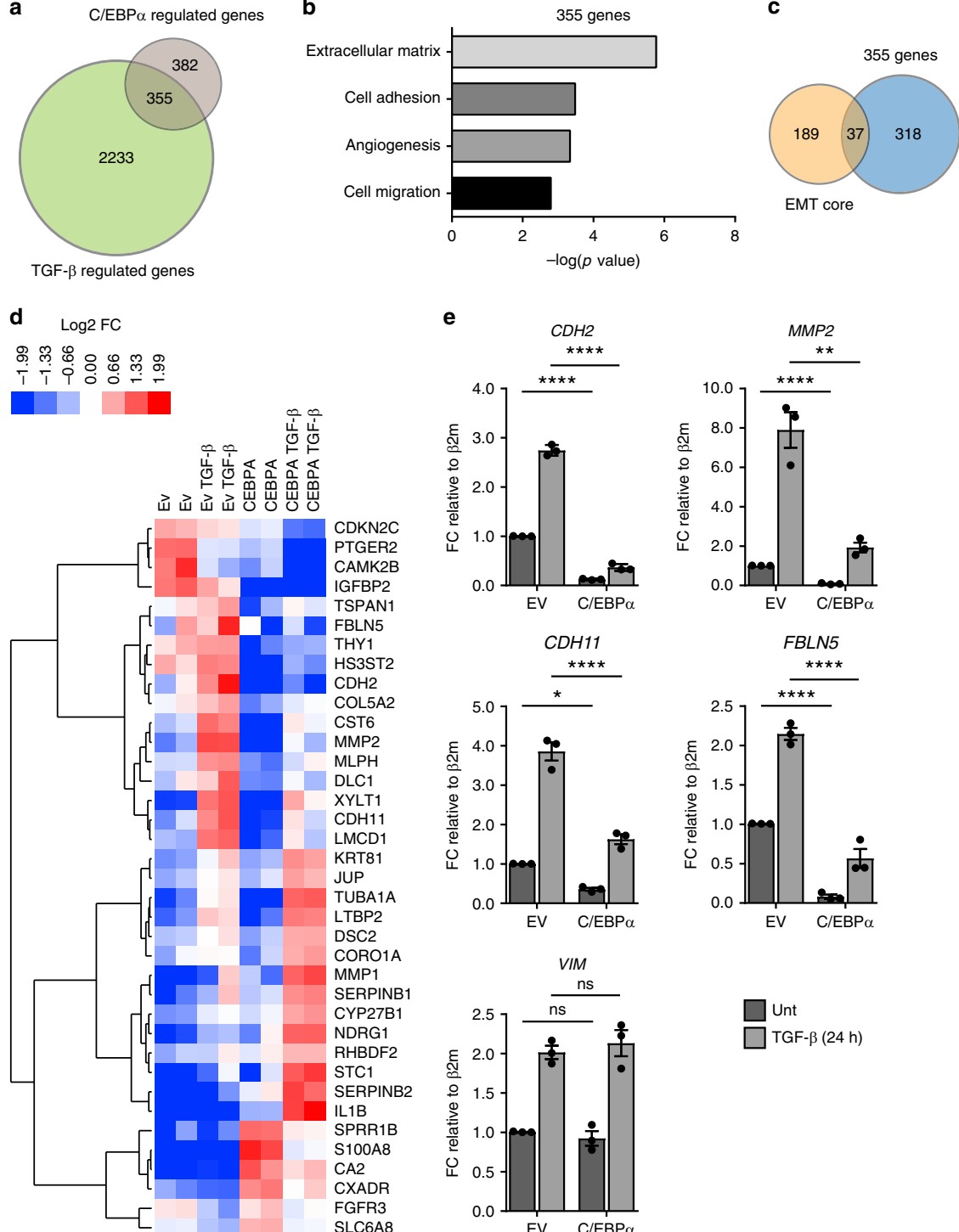

**Fig. 3 C/EBPa is important for TGF-β-induced EMT. a** HMLE cells expressing empty vector (EV) or constitutive *CEBPA* were either left untreated or treated with 2.5 ng/ml of TGF-β for 24 h. RNA samples were isolated, and purified RNA was sequenced as described in "Methods". Venn diagram showing the overlap of genes regulated by TGF-β with genes regulated by constitutive expression of C/EBPα. **b** Gene-ontology (GO) analysis on the 355 TGF-β- and C/EBPα-regulated genes. **c** Venn diagram showing the overlap of 355 TGF-β- and C/EBPα-regulated genes with public available EMT core genes. **d** Heatmap displaying gene expression of the 37 genes overlapped between EMT core and TGF-β-C/EBPα core. **e** qRT-PCR analysis validating RNA-sequencing data. HMLE cells expressing empty vector (EV) or constitutive *CEBPA* were left untreated or treated with 2.5 ng/ml of TGF-β for 24 h. Vimentin (*VIM*) was taken along as negative control. Data represented as mean ± SD of three independent experiments. *p*-values were calculated using unpaired two-tailed Student's *t* test. \**p* < 0.05, \*\**p* < 0.01, \*\*\**p* < 0.001, \*\*\*\**p* < 0.0001. ns not significant, FC, fold change, β2m beta-2-microglobulin.

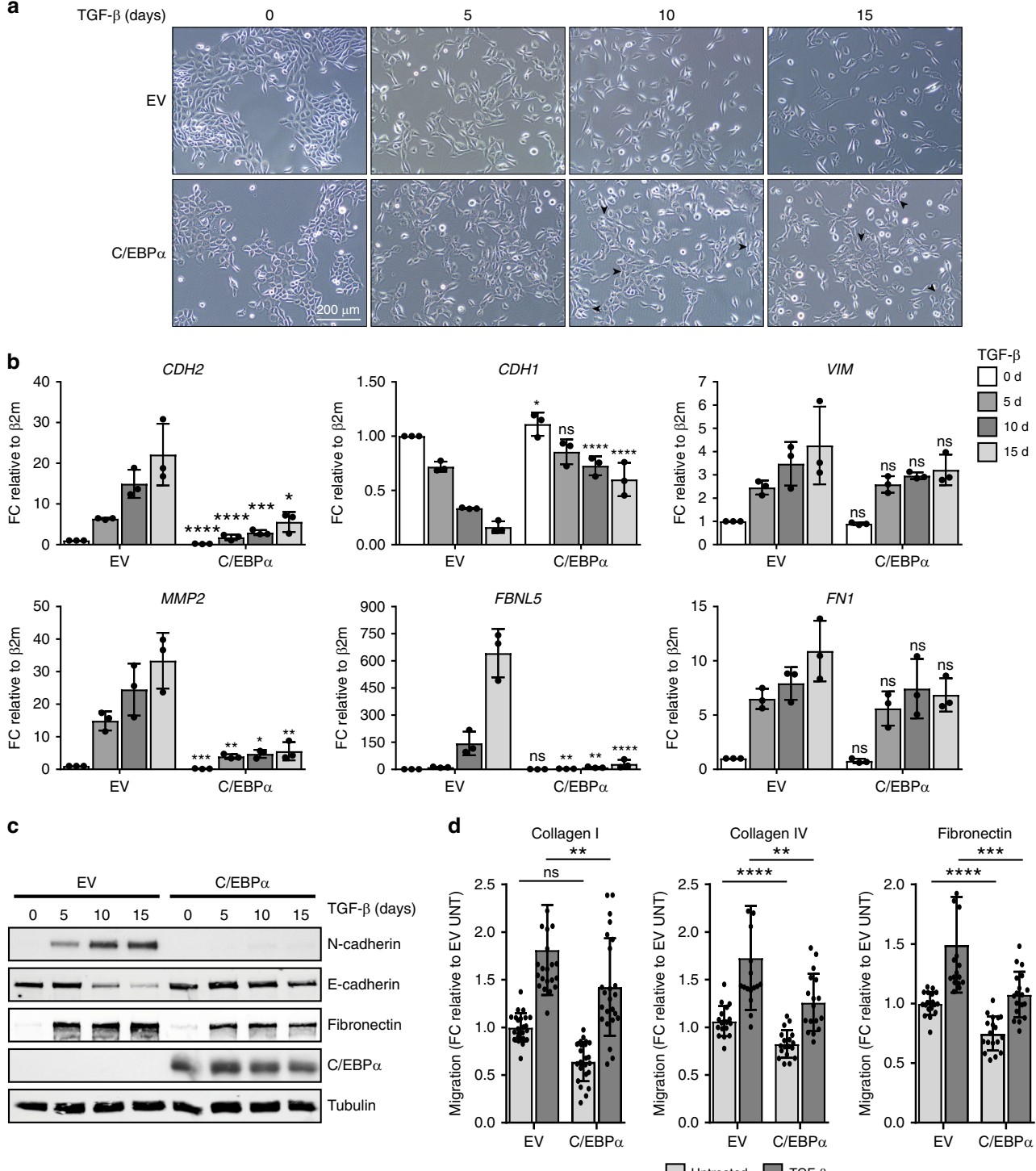

**Fig. 4 Constitutive expression of C/EBPα impairs TGF-β-induced EMT. a** HMLE cells expressing empty vector (EV) or constitutive *CEBPA* were either left untreated or treated with 2.5 ng/ml of TGF-β for 0, 5, 10, or 15 days. Bright-field microscopy images showing cell morphology during treatment. Data are representative of three independent experiments. Scale bar: 200 μm. **b** qRT-PCR results displaying the expression levels of well-stablished EMT markers during TGF-β stimulation in the presence of *CEBPA* constitutive expression. Data represented as mean ± SD of three independent experiments. *p*-values were calculated using unpaired two-tailed Student's *t* test. \**p* < 0.05, \*\**p* < 0.01, \*\*\**p* < 0.001, \*\*\*\**p* < 0.0001. ns not significant. **c** Immunoblotting results showing the protein levels of well-characterized epithelial and mesenchymal markers during TGF-β signaling in the presence of *CEBPA* constitutive expression. Data are representative of three independent experiments. **d** Quantification of wound closure in HMLE cells expressing EV or C/EBPα in untreated and TGF-β treatment conditions using 24-wells coated with the indicated substrates. Data represented as mean ± SD of *n* = 18–24 independent biological replicates. *p*-values were calculated using unpaired two-tailed Student's *t* test. \*\**p* < 0.01, \*\*\**p* < 0.001, \*\*\*\**p* < 0.0001. ns not significant, d days, FC fold change, β2m beta-2-microglobulin, UNT untreated.

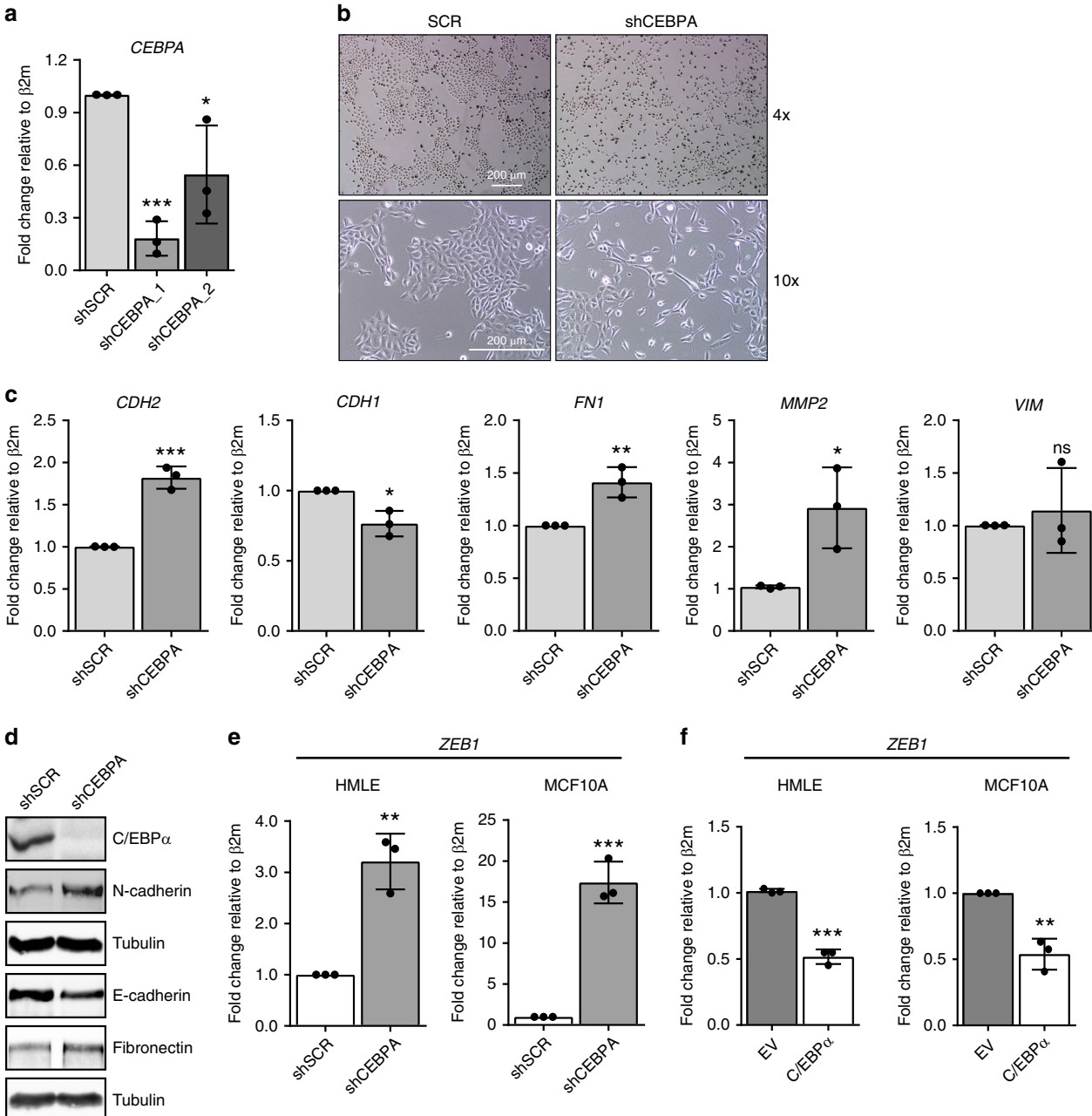

**Fig. 5 Knockdown of C/EBPα impairs epithelial homeostasis. a** The results from qRT-PCR analysis showing *CEBPA* mRNA levels in HMLE cells expressing shRNA control (SCR) or HMLE cells expressing two independent shRNAs targeting *CEBPA* (shCEBPA_1 and shCEBPA_2). Data represented as mean ± SD of three independent experiments. *p*-values were calculated using unpaired two-tailed Student's *t* test. \**p* < 0.05, \*\*\**p* < 0.001. **b** Bright-field microscopy images of HMLE cells expressing shRNA control (SCR) or shRNA targeting *CEBPA* in two different magnifications. Data are representative of three independent experiments. Scale bar: 200 μm. **c** qRT-PCR data showing the expression levels of known epithelial and mesenchymal markers upon *CEBPA* knockdown in HMLE cells. Data represented as mean ± SD of three independent experiments. *p*-values were calculated using unpaired two-tailed Student's *t* test. \**p* < 0.05, \*\**p* < 0.01, \*\*\**p* < 0.001. ns not significant. **d** The results from immunoblotting analysis displaying protein levels of well-characterized EMT markers upon *CEBPA* knockdown in HMLE cells. Data are representative of three independent experiments. **e** The results from qRT-PCR analysis showing the mRNA levels of transcription factor *ZEB1* upon *CEBPA* knockdown in HMLE and MCF10A cells. Data represented as mean ± SD of three independent experiments. *p*-values were calculated using unpaired two-tailed Student's *t* test. \*\**p* < 0.01, \*\*\**p* < 0.001. **f** *ZEB1* mRNA levels in HMLE or MCF10A cells expressing EV or constitutive *CEBPA*. Data represented as mean ± SD of three independent experiments. *p*-values were calculated using unpaired two-tailed Student's *t* test. \*\**p* < 0.01, \*\*\**p* < 0.001. β2m beta-2-microglobulin.

expression of E-cadherin in the absence of C/EBPα expression (Fig. 5d). Furthermore, we observed similar phenotypical and molecular changes in MCF10A cells upon shRNA-mediated depletion of *CEBPA* (Supplementary Fig. 4b–e). In addition to EMT-associated morphological and molecular changes, we also observed that depletion of *CEBPA* is sufficient to increase migration of MCF10A cells in a transwell assay system (Supplementary Fig. 4f). This is in accordance with our RNA-seq results that show that genes regulated by C/EBPα were associated with cell migration (Fig. 3b). To determine whether

reduced *CEBPA* expression was associated with increased expression of EMT-inducing transcription factors, expression levels of well-characterized EMT master regulators were evaluated. *CEBPA* knockdown resulted in robust induction of *ZEB1* expression in both HMLE and MCF10A cells (Fig. 5e), while no notable changes were observed in *SOX4*, *TWIST1/2*, or *SNAI2* (slug) expression (Supplementary Fig. 5a). Since decreased C/EBPα expression results in increased *ZEB1* levels, we assessed whether ectopic expression of C/EBPα conversely leads to a reduction in ZEB1 levels. qRT-PCR analysis of control and C/EBPα expressing HMLE or MCF10A cells revealed robust inhibition of *ZEB1* expression in the presence of C/EBPα (Fig. 5f). Bioinformatic analysis using the ContraV2 software revealed numerous highly conserved C/EBPα-binding motifs in the promoter region of *ZEB1* gene (Supplementary Fig. 5b), suggesting a repressor function for C/EBPα in the regulation of ZEB1. Taken together, these findings suggest that C/EBPα is indispensable for epithelial cells to retain their epithelial traits, and this may be, at least in part, due to repression of *ZEB1* expression.

**Restoration of C/EBPα levels induces EMT**. Since constitutive C/EBPα expression was sufficient to impair TGF-β-induced EMT, we evaluated whether restoration of C/EBPα expression in mesenchymal cells was able to drive mesenchymal-to-epithelial transition (MET). To this end, doxycycline (Dox)-inducible CEBPA-HMLE cells were generated, allowing conditional expression of C/EBPα. In order to induce mesenchymal features, control and CEBPA-inducible HMLE cells were first treated with TGF-β for 15 days in the absence of doxycycline. Subsequently, TGF-β and doxycycline were added for 60 h after which RNA and protein samples were isolated (Fig. 6a). As expected, doxycycline treatment showed a potent induction of C/EBPα expression in CEBPA-HMLE cells (Supplementary Fig. 6a). In addition, *CDH2*, *VIM*, *FN1*, and *ZEB1* mRNA expression were induced, and *CDH1* expression was repressed upon TGF-β treatment (Fig. 6b). The combination of TGF-β and doxycycline treatments in the control cells showed no differences in the regulation of gene expression compared with TGF-β-treated cells (Fig. 6b). However, TGF-β- and Dox-treated CEBPA-HMLE cells displayed strong reduction of the expression of mesenchymal markers and increased expression of the epithelial markers (Fig. 6b, Supplementary Fig. 6a). In accordance with qRT-PCR results, C/EBPα activation resulted in the inhibition of N-cadherin and fibronectin expression, and increased E-cadherin protein levels (Fig. 6c), supporting that conditional expression of C/EBPα in TGF-β-induced mesenchymal cells is itself sufficient to rescue their epithelial features. In order to assess whether C/EBPα was able to restore epithelial traits in a more complex epithelial cell culture system, we investigated the role of C/EBPα in MCF10A–epithelial cells cultured as three-dimensional (3D) spheroids. Culture of MCF10A cells on a reconstituted basement membrane, such as matrigel, results in the formation of well-organized epithelial spheroids comprising features found in human mammary gland in vivo[26]. TGF-β treatment has been shown to interfere with the architecture of MCF10A–epithelial spheroids, contributing to an active and invasive phenotype[27]. To confirm the effect of TGF-β signaling on the MCF10A mammary acini, cells were treated with TGF-β 2 days after plating. Four days of TGF-β stimulation resulted in the reduction of spheroid numbers and impairment of three-dimensional architecture (Fig. 6d). To assess whether expression of C/EBPα was sufficient to recover the formation of MCF10A–epithelial spheroids, we generated GFP-positive MCF10A cells expressing doxycycline-mediated

conditional activation of *CEBPA*. Prior to *CEBPA* induction, cells were treated with TGF-β for 48 h, in order to induce mesenchymal features (Fig. 6e). Subsequently, TGF-β was refreshed and doxycycline added for 48 h, after which cells were analyzed by fluorescence microscopy (Supplementary Fig. 6b). Spheroids were fixed and stained for C/EBPα and phalloidin (Fig. 6f). As expected, TGF-β treatment resulted in the disruption of the spheroids in both control and CEBPA-MCF10A cells. Likewise, the combination of TGF-β and doxycycline in empty-vector (EV) MCF10A cells resulted in the abrogation of epithelial spheroids. However, analysis of TGF-β and doxycycline treated CEBPA-MCF10A cells revealed the presence of well-structured spheroids, suggesting that activation of C/EBPα expression is sufficient to drive the reorganization of spherical architecture (Fig. 6f, g; Supplementary Fig. 6b). Immunoblotting analysis of control and CEBPA-MCF10A cells cultured in standard two dimensional system revealed that activation of C/EBPα contributes to decreased levels of N-cadherin and fibronectin whereas E-cadherin levels are maintained high (Supplementary Fig. 6c). Taken together, these data strongly suggest that restoration of C/EBPα expression in mesenchymal populations is sufficient to revert these cells toward an epithelial state, and this can be observed even under conditions of 3D epithelial organization.

**C/EBPα suppresses lung metastasis in a PyMT-mouse model**. To further study the potential for C/EBPα to modulate EMT-induced cancer progression in vivo, we ectopically expressed C/EBPα and empty-vector control in tumor organoids derived from YFP-positive PyMT breast tumors. This MMTV-PyMT; MMTV-Cre;R26R-YFP;E-cad-mCFP model of breast cancer metastasis has been previously shown to spontaneously develop ductal mammary tumors that metastasize primarily to the lungs with resemblances of EMT-mediated tumorigenesis[4,28]. Unlike breast cancer cell lines, tumor organoids display cellular heterogeneity, contributing to a more relevant model for development of breast cancer. Furthermore, while our previous analyses have focused on TGF-β-driven EMT, this PyMT model allows us to evaluate the impact of C/EBPα during EMT in an alternative system. As determined by qRT-PCR, *CEBPA* expression was effectively increased in C/EBPα-tumor organoids compared with control organoids (Supplementary Fig. 7a). To test whether ectopic expression of C/EBPα affects primary tumor growth, identical number of organoids expressing empty-vector (EV) control or C/EBPα were injected in five mice per group, and tumor size of both experimental groups was monitored during 5 weeks after injection. Analysis of primary tumor growth showed similar growth rate for both groups (Fig. 7a). Mammary tumors from this model, and similar to human ductal carcinomas, consist of a very small percentage of E-cadherin[low] cells which makes detecting changes in E-cadherin[low] cell number challenging[4]. However, we analyzed the percentage of E-cadherin[low] and E-cadherin[high] cells comprising the primary tumors of mice injected with control tumor organoids or tumor organoids overexpressing C/EBPα. Although not statistically significant, we observed that primary tumors derived from C/EBPα-expressing tumor organoids show a trend toward lower percentage of E-cadherin[low] cells compared with control group (Fig. 7b). Characterization of E-cadherin[low] cells demonstrated that these cells display EMT features, including increased expression of N-cadherin and ZEB1, as well as increased motility traits[4]. In order to assess the effect of C/EBPα on formation of metastasis, primary tumors of both groups were resected and mice were maintained for an additional 3 weeks, after which lungs were collected and analyzed for the occurrence of

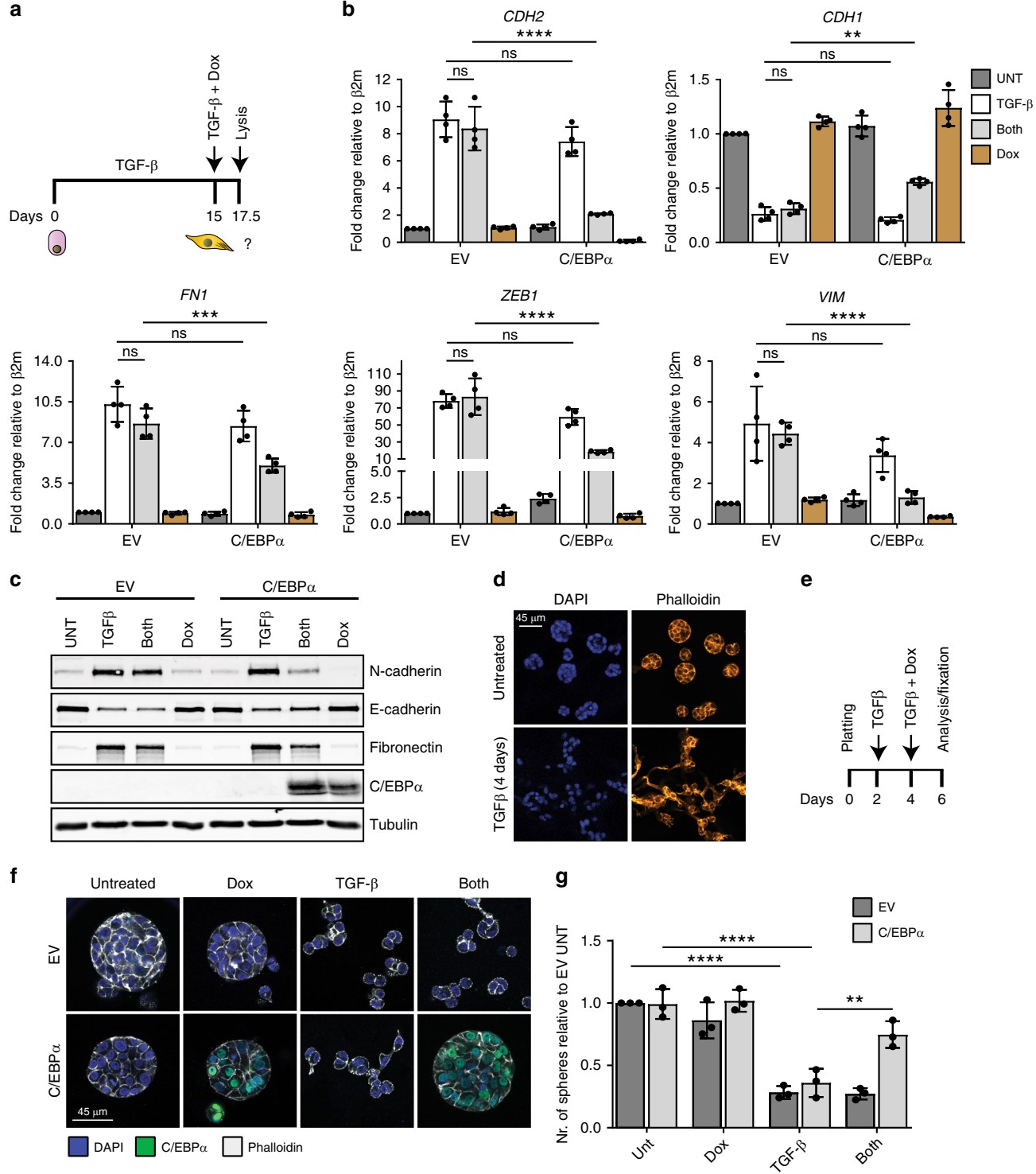

metastatic lesions. Macrometastasis in the lungs could readily be visualized by eye at the day of harvesting (Fig. 7c, arrowheads). Prior to collection, one EV-control mouse was found dead in the cage without obvious antemorten signs of illness, therefore its lungs were excluded for analysis. Comparison of the number of macrometastasis between the two groups revealed that mice injected with C/EBPα-overexpressing organoids (C/EBPα-mice) displayed significantly reduced numbers of metastatic lesions, suggesting that C/EBPα is sufficient to prevent tumor metastasis in vivo (Fig. 7c). Immunostaining confirmed that primary tumors of C/EBPα-mice expressed higher levels of C/EBPα

compared with control mice (Supplementary Fig. 7b). C/EBPα-positive cells could be detected in the surrounding tissue of the primary tumor of mice injected with control organoids, however no overlap with YFP-tumor tissue was observed, suggesting that some stromal cells or infiltrating cells endogenously express C/EBPα (Supplementary Fig. 7b, arrowheads). Immunohistochemistry analysis of various depths of lung tissue confirmed that mice injected with C/EBPα-overexpressing organoids display reduced number of metastatic lesions compared with control mice (Fig. 7d). Since there were still some metastases present in the C/EBPα-mice, it is relevant to evaluate whether

**Fig. 6 Restoration of C/EBPα rescues epithelial phenotype. a** Schematic representation of the experimental set up. HMLE cells were treated with 2.5 ng/ml of TGF-β for 15 days, and posteriorly C/EBPα expression was induced upon 200 ng/ml of doxycycline for 64 h. **b** The results from qRT-PCR showing the expression levels of known EMT markers in EV or CEBPA-inducible HMLE cells treated with TGF-β, doxycycline or both. Data represented as mean ± SD of four independent experiments. p-values were calculated using two-way ANOVA with Tukey's multiple comparisons test. **p < 0.01, ***p < 0.001, ****p < 0.0001. ns not significant. **c** Immunoblotting analysis of the expression levels of well-stablished EMT markers in TGF-β-treated HMLE cells in the absence or presence of C/EBPα overexpression. Data are representative of three independent experiments. **d** Confocal microscopy visualization of MCF10A–epithelial spheroids treated with 5 ng/ml of TGF-β for 4 days. Data are representative of three independent experiments. Scale bar: 45 μm. **e** Schematic representation of the 3D spheroid-formation assay. MCF10A cells expressing empty vector (EV) or doxycycline-inducible C/EBPα were stimulated with TGF-β (5 ng/ml), doxycycline (200 ng/ml), or both as indicated. **f** Confocal microscopy visualization of epithelial spheroids using MCF10A cells expressing empty vector or doxycycline-inducible C/EBPα in the presence of TGF-β, doxycycline (DOX), or both. Cells were stained for C/EBPa (green) and phalloidin (white). DAPI was used to visualize the nucleus (blue). Data are representative of three independent experiments. Scale bar: 45 μm. **g** Quantification of the number of spheres formed in MCF10A cells expressing empty vector or doxycycline-induced C/EBPα upon the treatment of TGF-β, doxycycline, or both. The number of spheres are relative to empty vector untreated. Data are represented as mean ± SD of three independent experiments. p-values were calculated using two-way ANOVA with Tukey's multiple comparisons test. **p < 0.01, ****p < 0.0001. β2m beta-2-microglobulin, EV empty vector, UNT untreated.

these lesions were positive for C/EBPα expression. Nuclear C/EBPα staining was absent in the YFP-positive metastatic lesions of EV-mice and greatly reduced in the metastasis of mice injected with C/EBPα-overexpressing organoids compared with its primary tumor tissue (Fig. 7e). This suggests that cells with the ability to metastasize are likely C/EBPα-negative/low. In order to confirm that reduction of metastatic potential observed in C/EBPα-expressing tumors was not related to the variability in tumor size, primary tumor of both groups was measured on the day of harvesting, and number of metastatic lesions were assessed (Supplementary Fig. 7c). Similarly, mice injected with C/EBPα-expressing organoids displayed a reduction in the number of metastatic lesions compared with control mice, supporting that C/EBPα expression impairs metastasis (Supplementary Fig. 7d). Taken together, these data suggest that in vivo, C/EBPα can function as a tumor suppressor in breast cancer, impairing metastasis, and thereby breast cancer progression.

## Discussion

EMT-inducing signals, such as the TGF-β pathway, contribute to the loss of epithelial features and acquisition of mesenchymal traits, and although it has been a subject of debate, several studies have reported that such cellular reprogramming impacts cell motility, extracellular matrix remodeling, cell extravasation, and chemotherapy resistance[4,5,29,30]. Whereas several studies have focused on understanding the role of the transcription factors induced during TGF-β-mediated EMT, knowledge regarding the regulation of epithelial transcription factors during this process is limited. Here, we demonstrate that repression of the transcription factor C/EBPα is essential for TGF-β-mediated EMT in human mammary epithelial cells. C/EBP transcription factors play a pivotal role during terminal differentiation of a variety of cells types, including mammary epithelial cells[12]. Decreased levels of C/EBPα have been found in several types of solid tumors, including breast cancer, however the mechanisms underlying this downregulation remain in general unknown[17]. By performing unbiased global RNA-sequencing analysis on TGF-β-treated HMLE cells, we identified C/EBPα expression as being one of the most repressed transcription factors, whereas no alterations in CEBPB expression were observed. These results contrast with previous observations that showed reduced CEBPβ expression upon TGF-β-mediated EMT in murine NMuMG cells[31]. These differences can be potentially explained due to species differences and/or context-dependent upstream signaling regulating C/EBPs expression[31].

TGF-β-mediated repression of C/EBPα expression was maintained during EMT. Since CEBPA mRNA levels were rapidly inhibited upon TGF-β treatment, we hypothesized that SMAD3 may be responsible for repression of CEBPA transcription.

Although SMADs are generally transcriptional activators, there are examples of genes that are transcriptionally repressed by this transcription factor family[32–34]. Similarly, we observed that occupancy of SMAD3 on CEBPA locus is enriched upon TGF-β treatment, and depletion of SMAD3 impaired TGF-β-mediated repression of CEBPA, supporting a role for SMAD3 as a transcriptional repressor of C/EBPα expression during TGF-β-induced EMT. Differences in the kinetics of C/EBPα mRNA and protein expression upon TGF-β treatment suggest that C/EBPα activity might also be regulated independently of protein degradation. We observed that short exposure to TGF-β induces cytoplasmic shuttling of C/EBPα protein in HMLE cells, suggesting that C/EBPα is posttranscriptionally regulated during TGF-β-mediated EMT, and together with its transcriptional repression enables a rapid induction of the EMT program.

Rapid reduction of C/EBPα expression during EMT appears to be critical to allow the progression of this program. Constitutive expression of C/EBPα during the initial 24 h of TGF-β stimulation was sufficient to repress genes important for the mesenchymal transition, including CDH2, MMP2, CDH11, and FBLN5. Proteins encoded by these genes regulate cell motility, extracellular matrix degradation, and transendothelial migration[35–38]. Indeed, C/EBPα overexpression was sufficient to impair TGF-β-induced cell migration. In addition, constitutive expression of C/EBPα during TGF-β-driven EMT impaired this program, as showed by the stable inhibition of crucial mesenchymal markers, including N-cadherin and fibronectin, and maintenance of high levels of epithelial marker E-cadherin. Induction of N-cadherin expression plays a pivotal role in EMT, and has been shown to be sufficient to promote cell migration and invasion in several cancer cell lines, regardless of E-cadherin expression[39,40]. In addition, mice expressing mutant N-cadherin die on day 10 after gestation due to impairment in heart formation, in which EMT is a critical process[41]. Therefore, repression of N-cadherin is essential to impair EMT and cancer progression. Furthermore, higher levels of E-cadherin were observed in TGF-β-treated C/EBPα-HMLE cells compared with the control cells. In addition to its role in preserving cell–cell/cell–basal membrane adhesion and cell polarity[41], studies showed that transmembrane E-cadherin-binding β-catenin is important to prevent nuclear translocation of β-catenin, thus preventing regulation of genes involved in promoting cell survival, proliferation, migration, and angiogenesis[41–44]. Taken together, by contributing to the repression of N-cadherin and maintenance of E-cadherin expression, C/EBPα is a critical negative regulator of TGF-β-induced EMT. Overexpression of C/EBPα in lung adenocarcinoma cells or enhanced C/EBPα expression by short-activating RNAs (saRNA) in hepatocellular carcinoma was observed to induce similar phenotypes[45,46]. This suggests that C/EBPα plays an

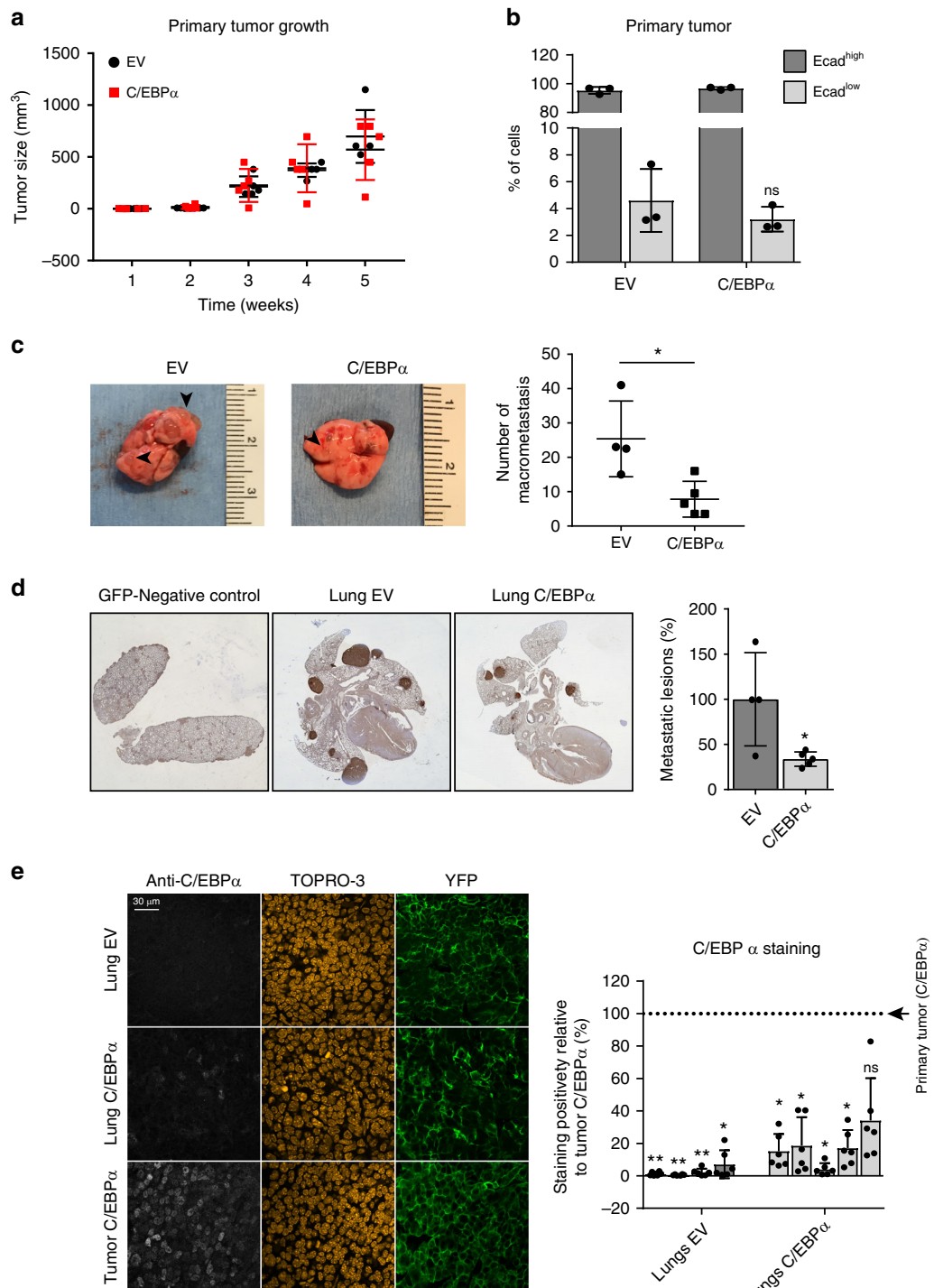

important and conserved role in preventing unwarrant EMT in multiple tissues.

Since C/EBPα depletion evoked features reminiscent of spontaneous EMT, we hypothesized that C/EBPα may act as a transcriptional repressor of a master regulator of EMT. The expression of EMT-inducers, including Snail, Twist, or Zeb1 transcription factors, can differentially contribute to the EMT program depending on the cell, tissue type and the signaling pathway triggering EMT[29]. Depletion of *CEBPA* in both HMLE and MCF10A cells resulted in strong upregulation of *ZEB1*, whereas no significant changes in other EMT factors were observed. The Zeb1 transcription factor acts as both transcriptional activator and

repressor by binding regulatory sequences at E-boxes. In several cellular contexts, TGF-β has been shown to activate Zeb1 resulting in the repression of E-cadherin and induction of N-cadherin and MMPs expression, whereas no major alterations in vimentin expression was observed[9]. In accordance with these observations, constitutive expression of C/EBPα during TGF-β-mediated EMT contributed to repression of ZEB1 (Supplementary Fig. 3b), N-cadherin and MMP2 as well as maintenance of E-cadherin levels, however no significant repression of vimentin was detected (Fig. 4b). These data, together with analysis of *ZEB1* regulatory region showing highly conserved C/EBPα-binding motifs, strongly suggests that C/EBPα contributes to epithelial homeostasis and

**Fig. 7 Ectopic C/EBPα expression in breast tumor organoids inhibits lung metastasis.** Tumor organoids derived from *MMTV-PyMT;MMTV-Cre;R26R-YFP; E-cad-mCFP* mice were transduced with EV control or C/EBPα-overexpressing construct and injected in NGS mice. **a** Tumor volume of primary mammary tumor. *n* = 5 per experimental group. *p*-values were calculated using two-way ANOVA with Tukey's multiple comparisons test (not significant). **b** Primary tumors of mice injected with control or C/EBPα-overexpressing tumor organoids (*n* = 3 per group) were dissociated to single cells, stained for extracellular E-cadherin, and analyzed for YFP, CFP, and E-cadherin expression by flow cytometry analyses. YFP-positive and DAPI-negative cells were further analyzed for low and high E-cadherin expression using both CFP and extracellular E-cadherin expression. Statistical significance was calculated using unpaired two-tailed Student's *t* test (ns not significant). **c** Representative lungs obtained from mice injected with EV or C/EBPα-overexpressing organoids (*n* = 4–5 per group). Arrowheads indicate an example of macrometastasis observed in the lungs. Quantification of the number of macrometastasis visible on the surface of the lungs from control and C/EBPα-mice is displayed. Data represented as mean ± SD. *p*-values were calculated using unpaired two-tailed Student's *t* test. *$p < 0.05$. **d** GFP staining of the metastatic lungs and quantification of the number of metastatic lesions present in various depths of the lung tissue. Data represented as mean ± SD of 4–5 mice per group. *p*-values were calculated using unpaired two-tailed Student's *t* test. *$p < 0.05$. **e** Immunofluorescence staining and quantification of C/EBPα expression in sections of the metastatic lungs of both experimental groups compared with primary tumor tissue derived from mice injected with C/EBPα-overexpressing organoids (4–5 mice per group). Scale bar: 200 μm. Each column represents the average value for each individual mouse relative to the positive control (primary tumor tissue developed from injected C/EBPα-overexpressing organoids). Six different sections were quantified per mouse. Data represented as mean ± SD. *p*-values were calculated using unpaired two-tailed Student's *t* test. *$p < 0.05$, **$p < 0.01$. ns not significant.

impairment of TGF-β-induced EMT through repression of *ZEB1*. Future studies will reveal whether C/EBPα is a direct repressor of *ZEB1*, or if such modulation requires intermediate steps of regulation.

The occurrence of metastasis is unavoidable for a great majority of the cancer patients, and despite the significant advances in treatment, metastases are still the major cause of cancer-related deaths. Eradication of metastasis often fails due to chemotherapy resistance. Recent studies have shown that cells that have undergone EMT and display mesenchymal traits are more insensitive to chemotherapy treatment[3,47]. Therefore, identification of novel molecular mechanisms underlying mesenchymal-to-epithelial switch (also known as MET) may contribute to the development of future cancer therapeutics and increase success rate of cancer treatment. We demonstrated that conditional activation of C/EBPα in TGF-β-induced mesenchymal cells is sufficient to strongly reduce expression of mesenchymal markers and partially restored expression levels of E-cadherin. Likewise, in a more complex biological setting, conditional activation of C/EBPα was sufficient to induce reorganization of epithelial spheroid architecture. These data suggest that restoration of C/EBPα expression in mesenchymal cells is sufficient to promote MET, thereby potentially restoring sensitivity toward chemotherapy. Generation of small molecules able to reach metastatic regions and activate *CEBPA* expression would therefore be strong therapeutic candidates.

We have demonstrated that C/EBPα is a direct transcriptional target of SMAD3, and its repression by TGF-β signaling is pivotal for the progression of TGF-β-induced EMT. Therefore, C/EBPα expression levels represent an important determinant of breast cancer progression. Moreover, by using an established mice model of EMT-induced breast cancer, we validated the anti-metastatic role of C/EBPα in vivo. Although the vast majority of GFP-positive cells colonizing the lungs of C/EBPα-mice were negative for C/EBPα expression, some cells showed positive nuclear staining at metastatic regions. Nonetheless, we cannot exclude that C/EBPα is transcriptionally inactive in these cells. As discussed previously, regulation of C/EBPα activity by post-transcriptional modifications has been shown to impact C/EBPα transcriptional output in liver cancer[16,48]. Importantly, our observations are supported by clinical data of breast cancer patients where *CEBPA* expression is found higher in primary normal tissue compared with primary tumor and markedly lower in primary tumors with metastasis compared with primary tumor without metastasis (Supplementary Fig. 8a, b). These data support the notion that C/EBPα is an inhibitory factor for tumor metastasis formation in a clinical setting. Moreover, we have also evaluated C/EBPα expression levels in various breast cancer cell lines and this also showed lower C/EBPα levels in metastatic breast tumor cell lines, such as MDA-MB-231 and MDA-MB-436, in comparison with untransformed HMLE cells (Supplementary Fig. 8c, d).

Based on our data, we propose a model that TGF-β-activated SMAD3 in epithelial cells promotes loss of C/EBPα expression resulting in the transdifferentiation into mesenchymal cells (Fig. 8). Maintenance of epithelial traits by C/EBPα can be envisioned by its direct involvement in the transcriptional activation of *CDH1* (E-cadherin) and repression of *ZEB1* (Supplementary Fig. 8e; Fig. 8a, b) and/or by the activation of an intermediate player responsible for transcriptionally repressing pro-oncogenic genes, such as *CDH2* and *MMP2* (Fig. 8c). In addition, we cannot exclude that C/EBPα itself cannot directly bind to the regulatory regions of these genes promoting its repression, as C/EBP transcription factors are both activators and repressors[49] (Fig. 8d). Upon TGF-β pathway activation, nuclear SMAD3 can bind and repress *CEBPA*. Loss of C/EBPα expression permits the activation of EMT program (Fig. 8e).

Taken together, our findings demonstrate that C/EBPα plays a pivotal role in the maintenance of epithelial homeostasis of human mammary cells and restoration of C/EBPα expression impairs EMT and the development of metastasis, supporting the view that cells require plasticity between epithelial and mesenchymal state in order to metastasize[44].

## Methods

**Cell culture.** Non-transformed immortalized human mammary epithelial cells expressing hTERT and SV40 large T and small t antigens (classified as HMLE cells and kindly provided by Prof. Dr. Robert Weinberg) were cultured in the MEGM medium (cc-3150, Lonza): DMEM/F12 media (Invitrogen) (1:1) supplemented with insulin (10 μg/ml, Sigma), EGF (20 ng/ml, Peprotech), hydrocortisone (0.5 mg/ml, Sigma), and 1% penicillin–streptomycin (Invitrogen). Mesenchymal-like phenotype cell cultures were obtained by supplementing the normal culture medium with 2.5 ng/ml of TGF-β1 (R&D Systems, 240-B-010). MCF10A cells (kindly provided by Dr. Patrick Derksen) were cultured in the DMEM/F12 media (Invitrogen) supplemented with horse serum (5% final, Invitrogen) insulin (10 μg/ml, Sigma), EGF (20 ng/ml, Peprotech), hydrocortisone (0.5 μg/ml, Sigma), 1% penicillin–streptomycin (Invitrogen), and cholera toxin (100 ng/ml, Sigma).

**Generation of C/EBPα cell lines.** To generate HMLE cells expressing constitutive C/EBPα and respective control, HMLE cells were transduced with pLZRS eGFP retroviral construct containing the coding sequence of human *CEBPA* or empty vector, respectively. Retrovirus particles were produced by transfection of the retroviral packing cell line Phoenix-ampho with 10 μg of DNA using poly-ethylenimine (PEI; Polysciences). Twenty-four hours after transfection, medium was replaced by MEGM:DMEM/F12 media. Viral supernatants were collected 24 h after, and filtered through a 0.22-μm filter and added to the HMLE cells. Cells were expanded and sorted according to their GFP positivity using flow cytometer (BD Biosciences). Cells were expanded for 2 weeks after sorting before experiments. The expression of C/EBPα p42 isoform was confirmed by immunoblotting analysis.

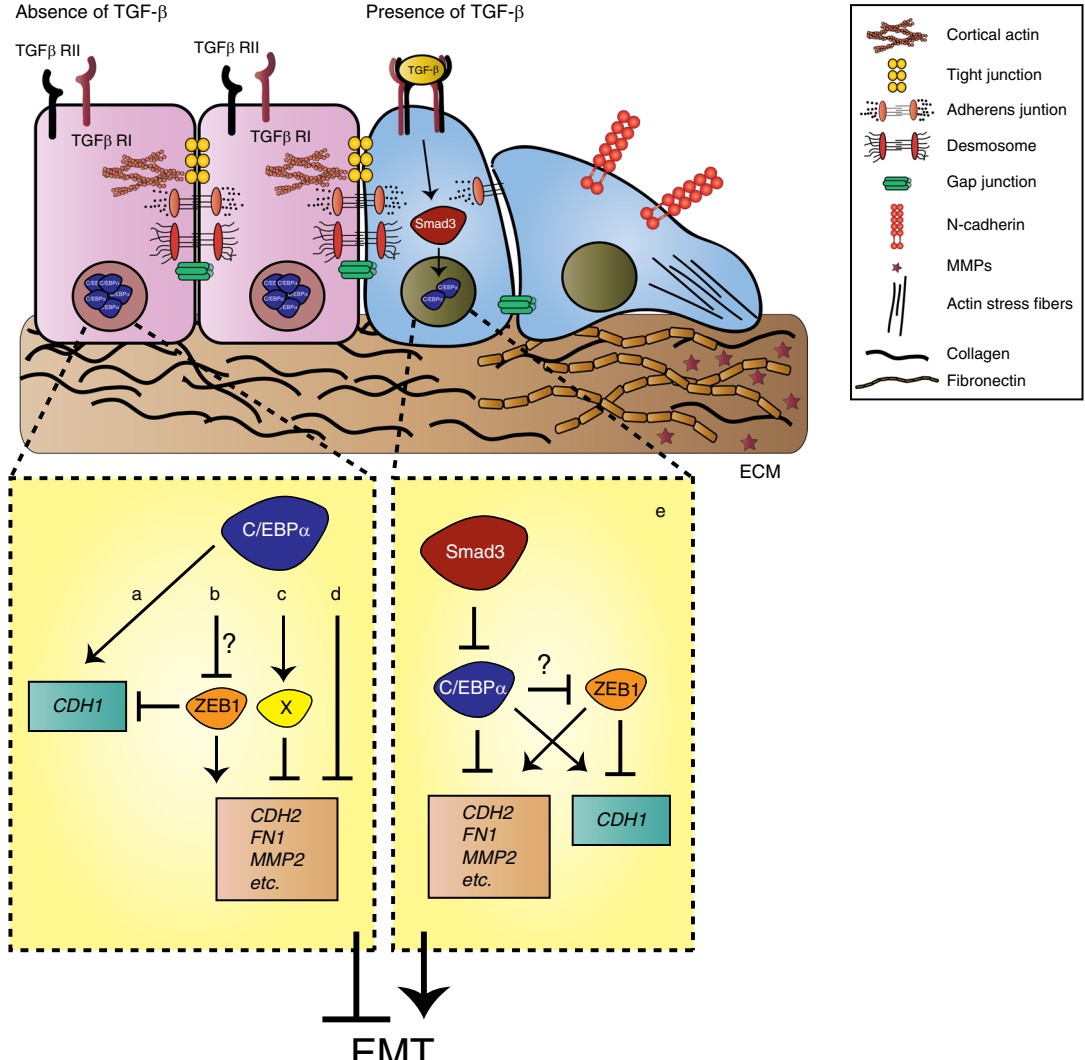

**Fig. 8 Schematic model for C/EBPα function during epithelial homeostasis and TGF-β-mediated EMT.** In non-transformed mammary epithelial cells, epithelial homeostasis is ensured by high levels of C/EBPα, which negatively regulates the expression of pro-mesenchymal proteins. C/EBPα-mediated repression of unwarranted EMT might be due to (**a**) direct transcription of epithelial proteins such as E-cadherin and (**b**) suppression of the EMT-inducer ZEB1. ZEB1 itself is able to directly repress *CDH1* (E-cadherin) expression and induce the transcription of several mesenchymal genes, including *CDH2* (N-cadherin) and *MMP2*. **c** Also, C/EBPα-mediated activation of an intermediate transcriptional repressor may be responsible for impairing EMT and/or (**d**) direct transcriptional repression by C/EBPα. Upon deregulation of TGF-β signaling, induction of EMT is induced by SMAD3-mediated inhibition of C/EBPα and (**e**) repression of EMT is alleviated.

To generate CEBPA-depleted HMLE cells and respective control, a pLKO.1-puro lentiviral construct expressing shRNA targeting *CEBPA* (shCEBPA_1: Sigma-Aldrich, TRCN0000007304 and shCEBPA_2: TRCN0000007306) or expressing shRNA control (Sigma-Aldrich, SHC002) were used, respectively. Lentivirus particles were produced by transfection of HEK293T cells with 3.25 μg of Pax2, 1.8 μg of pMD2.G, and 5 μg of shRNA constructs. Twenty-four hours after transfection, medium was replaced by MEGM:DMEM/F12 media. Viral supernatants were collected 24 h after, and filtered through a 0.22-μm filter and added to the HMLE cells. Cells were selected using 1 μg/ml of puromycin (invivoGen). Cells were maintained in 1 μg/ml of puromycin through the cell culture, except during experimental conditions where 0.3 μg/ml of puromycin was used.

To generate conditionally regulated C/EBPα cells, HMLE and MCF10A cells were transduced with a pINDUCER21 eGFP lentiviral construct containing the coding sequence of human *CEBPA* or empty vector as a control. Lentivirus particles were produced by transfection of HEK293T cells with 3.25 μg of Pax2, 1.8 μg of pMD2.G, and 5 μg of shRNA constructs. Twenty-four hours after transfection, medium was replaced by MEGM:DMEM/F12 media or DMEM/F12. Viral supernatants were collected 24 h after, and filtered through a 0.22-μm filter and added to the HMLE cells or MCF10A cells. Cells were expanded and sorted according to their GFP positivity using flow cytometer (BD biosciences).

Knockdown of *CEBPA* in HMLE cells using siRNAs was performed using 25 nM human CEBPA siRNA (Thermo Scientific, ON-TARGET plus SMARTpool) with Lipofectamine RNAiMAX reagents (Invitrogen), 72 h before harvesting samples.

All experiments in this paper have been carried out with pooled multiclonal populations.

**RNA sequencing.** HMLE cells were treated with TGF-β for 24 h after which RNA was isolated using the RNeasy Isolation kit (Qiagen) and according to the manufacturer's protocol. Purified RNA was subsequently repurified using mRNA-ONLY Eukaryotic mRNA Isolation Kit (Epicentre (Illumina, Inc.), Madison, WI, USA). Next, sequencing libraries were constructed using SOLiD Total RNA-Seq Kit (Applied Biosystems Life Technologies) according to the standard protocol recommendations for low input and sequenced on SOLiD Wildfire sequencer (Applied Biosystems Life Technologies) in a multiplexed way to produce 50-bp-long reads. Sequencing reads were mapped against the reference genome (hg19 assembly) using the BWA package. Only uniquely placed reads were used for further analysis. Cisgenome v2.0 software package was used to calculate reads per 1000 base pairs of transcript per million reads sequenced (RPKM) values for all RefSeq annotated genes. RPKMs were quantile normalized throughout all samples and presented as log2(RPKM) after adding small number to RPKM (0.1) to avoid

**Table 1 List of primers used for qRT-PCR.**

| Target | Sequence | Target | Sequence |
|---|---|---|---|
| hFN1 Fw | TGGCACCCCACGCTCAGATACA | hB2m Fw | ATGAGTATGCCTGCCGTGTGA |
| hFN1 Rv | CTCGCCAGGCAGGTTGACGG | hB2m Rv | GGCATCTTCAAACCTCCATG |
| hVIM Fw | ACCAACGACAAAGCCCGCGT | hCDH1 Fw | CACCACGTACAAGGGTCAGGTGC |
| hVIM Rv | CAGAGACGCATTGTCAACATCCTGT | hCDH1 Rv | CAGCCTCCCACGCTGGGGTAT |
| hTJP1 Fw | TGCGCTTACCACACTGTGATCCT | hSOX4 Fw | GTCCGCGCCTTGTACAGCGA |
| hTJP1 Rv | CCGACCATGGTTCAGGGGCA | hSOX4 Rv | GGCCTCGAGCTGGGAATCGC |
| hFBLN5 Fw | AATAAAACACCCGCGAGCCC | hSNAI1 Fw | TCCGGACCCACACTGGCGAGAA |
| hFBLN5 Rv | CACTGTGCCTGTGCATTCCC | hSNAI1 Rv | CCTGAGCAGCCGGACTCTTGGT |
| hCDH11 Fw | TCAAGGGCCCCAGAAATCAC | hSNAI2 Fw | CTGGGCGCCCTGAAGATGCAT |
| hCDH11 Rv | TTGAGCTCATCACGTCAGGG | hSNAI2 Rv | GGCTTCTCCCCCGTGTGAGTTCTA |
| hCDH2 Fw | AGTCACCGTGGTCAAACCAATCGA | hTWIST1 Fw | GCGTCGCCGCTCGAGAGATG |
| hCDH2 Rv | TGCAGTTGACTGAGGCGGGTG | hTWIST1 Rv | CGCTGTTGCTCAGGCTGTCGT |
| hMMP2 Fw | GGAGGCGCTAATGGCCC | hZEB1 Fw | AGCGCTTCTCACACTCTGGGTCTT |
| hMMP2 Rv | GGTATTGCACTGCCAACTCTTTGT | hZEB1 Rv | TGGCACCCACGTGCTCATTCG |
| hCEBPB Fw | CGACGAGTACAAGATCCGGC | hTWIST2 Fw | CGCCAGGGCTGTCCGTC |
| hCEBPB Rv | TGCTTGAACAAGTTCCGCAG | hTWIST2 Rv | CGGGTCTTCTGTCCGATGTC |
| hCEBPA 1 Fw | GCGGCGGCGGCGACTTT | hEPCAM Fw | CCATGTGCTGGTGTGTGAA |
| hCEBPA 1 Rv | GGTAGCCGGCGGCCGCGCA | hEPCAM Rv | TGTGTTTTAGTTCAATGAT |
| hCEBPA 2 Fw | TTGTATCTGGCCTCTGTGCC | hOCLN Fw | CCGAGTTTCAGGTGAATTGG |
| hCEBPA 2 Rv | GCCGACGGAGAGTCTCATTT | hOCLN Rw | GGAGTGTAGGTGTGGTGT |

log2 transformation of zero values. A fold change of value ≥ 1.5 (log2FC ≥ 0.58) and ≤ 0.72 (log2FC ≤ −0.48) were used as thresholds for differential expression analysis of the RNA-Seq data.

**Quantification of RNA expression (qRT-PCR).** mRNA was extracted from cells using the RNeasy Isolation Kit (Qiagen). According to the manufacturer's protocol for single-stranded cDNA synthesis, 500 ng of the total RNA was reverse transcribed using iScript cDNA synthesis kit (BIO-Rad). cDNA samples were amplified using SYBR green supermix (BIO-Rad), in a white 96-multiwell plate by Light-Cycler 96 system instrument (Roche) according to the manufacturer's protocol. To quantify the data, the comparative Ct method was used. Relative quantity was defined as $2^{-\Delta\Delta Ct}$ and β2-microglobulin was used as a reference gene. qRT-PCR primers are available bellow (Table 1):

**Immunoblotting analysis.** Cells were washed with PBS and lysed in Laemmli buffer (0.12 mol/L Tris-HCL (pH 6.8), 4% SDS, and 20% glycerol). Protein concentration was determined using Lowry protein assay. Equal amounts of sample (30–40 μg) were analyzed by sodium dodecyl sulfate polyacrylamide gel electrophoresis (SDS-PAGE) and electrophoretically transferred to the polyvinylidene difluoride membrane (Milipore). The membranes were blocked with 5% milk protein in TBST (0.3% Tween, 10 mM Tris pH 8.0, and 150 mM NaCl in $H_2O$) and probed with antibodies overnight at 4°. Immunocomplexes were detected using Odyssey imaging system (LI-COR Biosciences) or ECL (GE Healthcare), which were consequently exposed to Kodak XB films (Rochester). The following primary antibodies were used: anti-C/EBPα (Santa Cruz Biotechnology, sc-61), anti-N-cadherin (BD Biosciences, 610921), anti-E-cadherin (BD Biosciences, 610182), anti-Fibronectin (BD Biosciences, 610077), anti-α-tubulin (Sigma, T90026).

**Wound-healing migration assays.** Twenty-four-well plates were coated with collagen I, collagen IV, or fibronectin (all 50 μg/ml) for 2 h at 37 °C. In total, 200,000 control or C/EBPα-overexpressing HMLE cells were plated per well on the coated plates. When a confluent monolayer was achieved, migration was assessed by making a single straight scratch in the monolayer using a 20–200 μl pipette tip at the end of the day. Wells were washed twice with PBS to remove loose cells, and 0.5 ml of medium containing TGF-β (2.5 ng/ml) was added, followed by a 20 h incubation at 37 °C, 5% $CO_2$, and ambient oxygen level (20%). Images were recorded at $t = 0$ and $t = 20$ h at similar locations, after which the scratch surface was determined using ImageJ software to determine wound closure.

**Transwell migration assays.** Transwell assays were performed according to the manufacturer's protocol (CLS3464-12EA, Corning). Briefly, 30,000 scrambled control or CEBPA shRNA MCF10A cells were plated in the inner compartment of the transwell plate in 300 μl 1% FBS MCF10A medium, with 700 μl 10% FBS MCF10A medium in the outer compartment. The plate was subsequently incubated for 24 h at 37 °C and 5% $CO_2$. Subsequently, media was removed and cells were washed, fixed, and stained with DAPI to quantify the number of migrated cells using ImageJ software.

**Confocal microscopy.** Cells were cultured on microscope glasses (Sigma-Aldrich). Coverslips were washed twice with PBS and fixed using PBS containing 4% paraformaldehyde (Merck) for 20 min at room temperature. Cells were washed twice with PBS and permeabilized with PBS containing 0.25% Triton (Sigma) for 5 min at room temperature. Posteriorly, cells were pre-incubated with PBS containing 2% of bovine serum albumin (Sigma) for 60 min at room temperature. After the blocking step, cells were incubated overnight at 4° with anti-C/EBPα (Abcam, ab128482), anti-E-cadherin (Abcam, ab1416), and anti-Fibronectin (BD Biosciences, 610077). Cells were washed twice in PBS, and respective secondary antibodies (BD Biosciences) were incubated for 60 min, at room temperature. Lastly, cells were washed twice in PBS and coverslips were mounted in Prolong Gold antifade reagent with DAPI (Invitrogen). Confocal images were acquired using a Zeiss LSM 700 fluorescence microscope (Zeiss).

**Chromatin immunoprecipitation—sequencing.** Chromatin immunoprecipitation was performed in HMLE cells using 5 of 15 -cm dishes per condition. HMLE cells were either treated with TGF-β (2.5 ng/ml; R&D Systems, 240-B-010) for 16 h or left untreated. Subsequently, double cross-linking was performed using Di(N-succinimidyl) glutarate (DSG) for 45 min followed by a 30 min incubation with formaldehyde. The reaction was quenched using incubation with 0.1 M glycine for 5 min, after which cells were washed in PBS and nuclear extracts were generated. Sonication was subsequently performed for 8 min at maximum output (ultra-sonicator, Covaris), after which immunoprecipitation was performed with 1 μg of the rabbit anti-SMAD3 (ab28379) antibodies coupled to protein A/G sepharose beads (Santa Cruz Biotechnology). Sequencing libraries were generated using the TruSeq LT kit (Illumina), and sequencing was performed on NextSeq platform (Illumina). Sequencing reads were mapped to the reference genome assembly (hg19) using Bowtie2, and peak calling was performed using MACS2. Data analysis was performed using the HOMER software package.

**3D matrigel assay.** Eight-well glass bottom chamber (Ibidi) was coated with 40 μl of cultrex matrigel (R&D Systems) for 1 h at 37°. MCF10A cells were plated (5000 cells per chamber) in MCF10A assay medium (DMEM/F12 media (Invitrogen) supplemented with horse serum (2% final, Invitrogen), insulin (10 μg/ml, Sigma), hydrocortisone (0.5 μg/ml, Sigma), cholera toxin (100 ng/ml, Sigma), and 1% penicillin–streptomycin (Invitrogen)[26]. At specific experimental points, 5 ng/ml of TGF-β was added. For immunofluorescence staining, medium was gently removed and cells were fixed using PBS containing 4% paraformaldehyde (Merck) for 15 min at room temperature. Cells were washed twice with PBS and permeabilized with PBS containing 0.20% Triton (Sigma), 1% dimethyl sulfoxide (DMSO; Santa Cruz Biotechnology), and 2% bovine serum albumin (Sigma) for 60 min at room temperature. After the permebealization/blocking step, cells were incubated overnight at 4° with anti-C/EBPα (Cell Signaling Technology, 2295). On the following day, cells were left for 30 min at room temperature and thereafter washed twice in PBS. Respective secondary antibody (BD Biosciences) and phalloidin (Cell Signaling Technology, 8940) were incubated for 2 h at room temperature. Lastly, cells were washed twice in PBS, and 15 μl of Prolong Gold anti-fade reagent with DAPI (Invitrogen) was added. Confocal images were acquired using a Zeiss LSM 700 fluorescence microscope (Zeiss).

**Cell growth analysis**. Equal numbers of control or C/EBPα-overexpressing HMLE cells were seeded on a tissue-culture-treated six-well plate. After 4 days, cell dissociation using trypsin was performed, and the number of cells in both experimental groups was analyzed using trypan blue stain.

**Luciferase assays**. For the luciferase assays, HEK293T cells at 50% confluency were transfected in 24-well plates with 0.1 μg of CDH1 promoter luciferase reporter (pGL3 CDH1) or empty-vector control (pGL3), with 0.1 μg of pcDNA3 empty vector or pcDNA3- HA-CEBPA and 0.02 μg pRL-TK Renilla (Promega) as a transfection control. The cells were lysed in 50 μl passive lysis buffer 3 days post transfection. The soluble fraction was subsequently assayed for luciferase activity with a Dual-Luciferase Reporter Assay System (Promega, USA).

**PyMT tumor organoids**. PyMT tumor organoids were generated from MMTV-PyMT;MMTV-Cre;R26R-YFP;E-Cad-mCFP mice on a FVB genetic background. Tumors were harvested and enzymatically digested using trypsin (from bovine pancreas, Sigma) and collagenase A (Roche). The digested tumors were spun down in several steps until only the cell fragments of 200 to 1000 cells were left. These organoids were embedded in BME (reduced growth factor basement membrane extract type 2, PathClear)[4,50]. Organoids were maintained in medium consisting of DMEM/F12 Glutamax supplemented with HEPES (1 M Gibco), 1% penicillin–streptomycin, FGF (Life technologies) and B27 (50× Gibco).

Lentiviral particles from pLEX307-C/EBPα overexpressing construct and pLEX307-empty-vector construct were produced using standard protocols and as detailed above. Viral supernatants were concentrated using a centrifugal filter unit (Milipore, UFC901024) for 60 min at 4°. The organoids were digested using Trypsin (Invitrogen Technologies), and mechanical disruption was used to generate small fragments of ~30–50 cells. The concentrated virus was diluted in DMEM/F12 Glutamax (Gibco) supplemented with HEPES (1 M Gibco), penicillin–streptomycin, 8 μg/ml polybrene (Sigma), and 10 μM Y27632 (Sigma). The organoids and virus-medium were spun in a 48-well plate at 600 rcf for 60 min at 32° and afterwards placed in a 37 °C, 5% CO$_2$ incubator for 3 h. Next, the organoids were washed and plated. The organoids were selected using puromycin (100 ng/ml) Life Technologies).

**Mice**. Non-obese diabetic SCID IL-2 receptor gamma chain knockout (NSG) mice (own colony) were housed under IVC conditions. Mice received food and water ad libitum. All experiments were carried out in accordance with the guidelines of the Animal Welfare Committee of the Royal Netherlands Academy of Arts and Sciences, The Netherlands.

**Organoid transplantation and mastectomy in mice**. C/EBPα-overexpressing organoids or empty-vector control organoids were harvested and made into smaller pieces using mechanical disruption. The organoids were diluted in PBS. NSG mice were sedated using isoflurane inhalation anesthesia (1.5% to 2% isoflurane/O$_2$ mixture), and the organoids were injected beneath the 4th nipple. If the mouse developed a tumor of 1500 mm$^3$, a mastectomy was performed while sedated using isoflurane inhalation anesthesia (1.5% to 2% isoflurane/O$_2$ mixture). All the mice were killed 5–8 weeks after injection. Metastatic lesions in the lungs were counted by two independent researchers.

**Mouse tumor and tissue processing for histology**. Part of tissues were fixed in periodate-lysine-paraformaldehyde (PLP) buffer (2.5 ml 4% PFA + 0.0212 g NaIO$_4$ + 3.75 ml L-Lysine + 3.75 ml P-buffer (pH 7.4) at 4 °C. The following day, the fixed tumors and tissues were washed twice with P-buffer and placed for at least 6 h in 30% sucrose at 4 °C. The tumors and tissues were then embedded in tissue freezing medium (Leica Microsystems) and stored at −80 °C before cryosectioning. Other part of the tissues were fixed in 4% paraformaldehyde, dehydrated, and embedded in paraffin.

**GFP staining of paraffin sections**. Paraffin sections of 4 μm were firstly immersed in xylene for 10 min at room temperature and subsequently immersed in different percentages of ethanol (100%, 95%, 70%, and 50%) for ~2 min each step. Tissue was rinsed in deionized water and boiled in citrate buffer (10 mM C$_6$H$_5$Na$_3$O$_7$.2H$_2$O pH 6.0) for 20 min. Slides were put aside to cool down and thereafter washed in PBS. After that, slides were blocked in PO-block buffer (95 ml PBS with 5 ml 30% H$_2$O$_2$ (37%)) for 15 min at room temperature. Washing step in PBS was performed, and second block in PBS containing 2.5% bovine serum albumin (Sigma) for 30 min at room temperature was followed. Subsequently, tissue was incubated with anti-GFP antibody (Abcam, ab6673) in PBS containing 0.5% bovine serum albumin (Sigma) for 3 days at 4°. After primary antibody incubation, tissues were washed in PBS and incubated with rabbit anti-goat antibody (BIO-RAD, STAR194) for 1 h at room temperature. After that, samples were washed in PBS and incubated with HRP-labeled anti-rabbit (ImmunoLogic, DPVB110HRP) for 30 min at room temperature. Tissues were washed in PBS and incubated with 3,3' diaminobenzidine (DAB) containing 0.1% of H$_2$O$_2$ (from a 30% stock) for 10 min at room temperature. Washing steps in PBS were followed, and tissues were incubated with hematoxyline counterstaining for 5 min at room temperature. Lastly, samples were immersed in different percentages of ethanol (50%, 70%, 95%, and 100%) for ~2 min each step, followed by an incubation with xylene for 10 min at room temperature and subsequently mounted with pertex. Tissues were analyzed with an inverted bright-field microscope at 10 and 40 magnifications to morphologically identify metastasis. Metastatic lesions in the lungs were counted by two independent researchers.

**Immunofluorescence staining of frozen tissue sections**. Frozen tissues sections of 16 μm were rehydrated in 0.1 M Tris pH 7.4 for 10 min at room temperature and subsequently permeabilized in 0.1 M Tris pH 7.4 containing 0.25% Triton (Sigma) for 5 min at room temperature. Posteriorly, blocking was performed in 0.1 M Tris pH 7.4 containing 2% bovine serum albumin for 45 min at room temperature. After blocking step, tissues were incubated overnight at 4° with anti-C/EBPα (Cell Signaling Technology, 2295). In the following day, samples were washed twice in 0.1 M Tris pH 7.4, and incubated with donkey anti-rabbit secondary antibody Alexa Fluor 555 (Thermo Scientific, A31572) and TOPRO-3 for 2 h at room temperature. Lastly, tissues were washed twice in 0.1 M Tris pH 7.4 and mounted using vectashield anti-fade mounting medium (Vector labs). Confocal images were acquired using a Zeiss LSM 700 fluorescence microscope (Zeiss) at 63 magnification.

**Flow cytometry on mouse material**. Tumors were collected and minced manually on ice using sterile scalpels. The tumor mass was digested in PBS supplemented with 25 μg/ml DNase I (Roche) and 5 Wünsch units TH Liberase /ml (Roche) at 37 °C for 35 min, followed by mashing through a 70-μm filter (BD Falcon) while adding DMEM/F12 + GlutaMAX (GIBCO, Invitrogen Life Technologies) supplemented with 5% (v/v) fetal bovine serum (Sigma), 100 μg/ml streptomycin, and 100 U/ml penicillin (Invitrogen Life Technologies), 5 ng/ml insulin (I0516-5ML Sigma, St. Louis, MO, USA), 5 ng/ml EGF (Invitrogen), and 25 μg/ml DNase. After spin down (4 min at 500 RCF at RT), the pellet was resuspended in 6 ml 5 mM EDTA/PBS, after which a Ficoll gradient (Histopaque-1077, Sigma) was used to select for live cells (30 min at 400 RCF at RT, break 1). Cells were washed once in 5 mM EDTA/PBS and centrifuged (4 min at 500 RCF at RT) before proceeding with antibody labeling. The cells were blocked in 80% FACS buffer (5 mM EDTA in PBS supplemented with 5% fetal calf serum)/20% serum mix (50/50 normal goat serum (monx10961, Monosan) and FcyII/III receptor blocking serum 2.4G2 (kind gift from Kiki Tesselaar, UMCU, The Netherlands)) for 10 min on ice before labeling with E-cad-eFluor660 (DECMA-1, eBioscience). The cells were analyzed on FACS Jazz (BD biosciences) using the following strategy: a broad FSC SSC gate was followed by a gate excluding doublets, after which immune cells and megakaryocytes were excluded in a dump channel. YFP-positive and DAPI-negative tumor cells were gated using E-cad-mCFP and E-cadherin 647 lasers.

**Statistical analysis**. Data represented as mean ± SD of at least three independent experiments. Differences were analyzed by unpaired two-tailed $t$ test between two groups and by two-way ANOVA for differences between more than two groups. Exceptionally, qPCR results obtained on Fig. 2b were analyzed using unpaired one-tailed $t$ test. ns indicates nonsignificant. *$p < 0.05$; **$p < 0.01$; ***$p < 0.001$; ****$p < 0.0001$.

**Reporting summary**. Further information on research design is available in the Nature Research Reporting Summary linked to this article.

## Data availability

Data presented on Fig. 2a and S1c were generated by analyzing the data available under the accession numbers GSE104761 and GSE24202, respectively. The data sets generated during this study (RNA-seq) were deposited under the accession number GSE143612. Uncropped western blot images can be found in the Supplementary Information section (Supplementary Figs. 9–15).

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

## Acknowledgements

We would like to thank Prof. Robert A. Weinberg and Dr. Patrick Derksen for providing us the HMLE and MCF10A cells, respectively. A.R.L. was supported by a FCT (Fundação para a Ciência e a Tecnologia) fellowship. A.R.L., S.J.V., M.G.R. and C.L.F. were supported by a grant from the Dutch Cancer Society (KWF).

## Author contributions

P.J.C. supervised the study. A.R.L., M.G.R., D.S., J.v.R. and P.J.C. designed experiments. A.R.L., C.L.F., C.E.P., A.S.M. and D.S. performed experiments. S.J.V. performed NGS analysis. A.R.L., M.G.R. and P.J.C. interpreted experiments. A.R.L. and P.J.C. wrote the paper.

## Competing interests

The authors declare no competing interests.
