## [Peer Review File · Nature Communications]

Reviewers' comments:

Reviewer #1 (Remarks to the Author):

In the manuscript "C/EBP α is (a) crucial determinant of epithelial homeostasis by preventing epithelial-to-mesenchymal transition", the authors set out to identify transcription factors suppressed during the EMT process in human mammary cells. They find C/EBP α expression to be suppressed in response to TGF- β and show a negative relationship between expression of C/EBP α and several mesenchymal markers, including the transcription factor ZEB1, in two different mammary epithelial culture cell lines. Conversely, they find a positive relationship between expression of C/EBP α and the epithelial protein, E-cadherin. They go on to demonstrate that cells that have undergone a TGF- β -induced EMT reverted to an epithelial-like state upon C/EBP α re-expression under both 2D and 3D culture conditions. Finally, C/EBP α expression in the primary tumor prevented the formation of metastasis in an orthotopic mouse model. The authors conclude, "that C/EBP α is required for maintaining epithelial homeostasis" and "that C/EBP α is a master epithelial "gatekeeper" whose expression is required to prevent unwarranted mesenchymal transition". The manuscript provides data supporting a negative relationship between levels of C/EBP α expression and induction of EMT, and supporting the idea that C/EBP α is able to suppress EMT. As I outline below, the manuscript in its current form requires significant revision, it lacks clearly stated rationale for several key experiments and provides insufficient data to support several key conclusions.

Major comments

1. In general the manuscript would greatly benefit from stronger rationale for the experiments conducted. Examples include: the selection of SMAD3 as a potential repressor of C/EBP α , is SMAD2 a poor candidate; the focus on C/EBP α instead of E2F2 or E2F8, C/EBP α itself has been shown to inhibit the cell cycle in myeloid cells; the use of PyMT model which has been shown to metastasize to the lung without undergoing EMT, Ref 4 (Fischer et al, Nature 2015).

2. C/EBP α mRNA levels appear to decrease by \sim 75% after 24h of TGF- β treatment (Fig 1d), this decrease coincides with a dramatic increase in N-cadherin and fibronectin protein levels (Fig 1e). Unlike C/EBP α mRNA, the amount of C/EBP α protein appears unchanged in Fig 1d and mildly decreased in Fig 2e (provide quantification of C/EBP α protein levels relative to loading controls). How do the authors reconcile the dramatic increase in mesenchymal markers with a relatively modest decrease in C/EBP α ? Fig 1f shows a dramatic relocalization of C/EBP α out of the nucleus at 15 days, does this also occur at early time points? Investigation of how C/EBP α protein localization is

regulated in response to TGF- β is likely to provide valuable insight. Does mutation of C/EBP α DNA binding or NLS disrupt its ability to suppress EMT?

3. To explain the early decrease in C/EBP α mRNA, SMAD3 was assessed for binding to the C/EBP α locus in a role as putative repressor. As discussed above, the modest decrease in C/EBP α protein in response to TGF- β after 24h questions the relevance of inhibiting C/EBP α mRNA expression by SMAD3. To determine if SMAD3 is able to repress CEBPA expression, knock-down studies were done using a single hairpin. Could this observation be due to off target effects, demonstrate the result in 2d and 2e using multiple hairpins. Data shown in Fig 2d and 2e indicate an increase in basal levels of CEBPA expression upon SMAD3 knock-down and a subsequent CEBPA decrease in response to TGF- β , both at the mRNA and protein levels. The decrease in Fig 2d is reported as not significant, however this appears to stem from an increase in experimental error. If the same level of standard deviation is achieved in the shSMAD3 group as shSCR would the reported conclusion stand? The basal increase in CEBPA may suggest a potential mechanism of long term suppression, however the immediate decrease in CEBPA mRNA in response to TGF β suggest other regulation. Also considering C/EBP α protein stability at 24h, another mechanism of C/EBP α mediated N-cadherin and Fibronectin suppression requires consideration.

4. ChIP-seq data indicated two regions where SMAD3 was enriched in proximity to the CEBPA locus. Enrichment was validated using ChIP-qPCR. To improve robustness of the validation experiment, consider the addition of a TGFBR1/ALK5 inhibitor (eg SB431542) to limit basal SMAD3 activation/nuclear accumulation in the control. Do the two regions identified contain SMAD3 binding sequence motifs? Is SMAD3 binding these regions directly?

5. TGF- β is used to induce EMT throughout the manuscript. Does overexpression of C/EBP α (as shown in Fig 3e) affect SMAD2/3 phosphorylation at 1 and 24 hrs after TGF- β treatment? Could the decrease in TGF- β response reported in Fig 3e be due to interference with the cell's ability to signal properly? What was the period of time between sorting of GFP positive cells and initiation of the experiment? Overexpression of C/EBP α appears to affect basal levels of several mesenchymal associated mRNAs. What are the relative levels of C/EBP α overexpression to endogenous protein/mRNA in Fig 3?

6. The authors state that in cancer, EMT is important for the acquisition of migratory and invasive properties and show that genes common to C/EBP α and TGF- β expression profiles are related to ECM remodeling and cell migration. Does overexpression/knock-down of C/EBP α affect migration and invasion of TGF- β stimulated epithelial cells in Boyden chamber assays?

7. The manuscript states that C/EBP α 1) is crucial to maintain epithelial architecture and 2) prevent EMT. The authors reach these conclusions without providing a mechanism showing how C/EBP α promotes expression of epithelial genes or limits expression of mesenchymal genes. A potential mechanism of ZEB1 transcriptional inhibition is highlighted in the results and discussion, however no data is provided to support such a mechanism. Evidence of such regulation in epithelial cells would be of great interest. Recent work in adipose cells has shown that ZEB1 is a positive regulator of C/EBP α and adipocyte differentiation (Gubelmann et al, Elife 2015), notably knockdown of ZEB1 greatly decreased C/EBP α in these cells. One might expect that C/EBP α must be regulated differently in epithelial cells undergoing EMT, where C/EBP α is decreased, as ZEB1 is increased. Can the authors comment on the regulatory divergence among cell types?

8. On page 16, the authors state, "C/EBP α is indispensable for epithelial cell to retain their epithelial traits". However, the sole epithelial marker analyzed, E-cadherin (in panel 5c) showed a ~20% decrease, while mesenchymal markers showed more pronounced changes. Is C/EBP α playing a direct role in maintaining expression of epithelial genes or is it more indirect, by preventing expression of mesenchymal transcription factors and genes, which in turn suppress epithelial traits?

9. Cells from PyMT tumors must extravasate, survive and intravasate into the lung tissue prior to formation of the metastases observed in Fig 7. Was blood drawn to determine if EV and C/EBP α tumors released equal number of invasive cells into the blood stream? Perhaps C/EBP α overexpression allows fewer cells to escape the primary tumor, a possibility considering the gene expression profile associated with C/EBP α . Is it possible to analyze the epithelial/mesenchymal status of these cells? This is important to address.

Minor comments and suggestions.

Provide table with the sequence of primers used in this study.

Title: 'epithelial homeostasis' evokes the role of C/EBP α in maintenance of epithelial tissues, which is not the focus of this paper. The manuscript focuses on its role in limiting EMT aided metastasis instead, consider revising.

Introduction: page 3, line 3rd from the end – "establishing" instead of "stablishing"

Results: First line on page 12, extra comma. First sentence on page 14, editing, delineate or define. Last line on page 14, last line – “difference” instead of “differences”

In the material and methods, indicate the concentration of cell media components.

Figure 1c is more appropriate for the supplement.

In supplementary fig. 1, figure 5 and 6 note the use of “,” (commas) instead of “.” (decimals).

There are formatting problems in the x axis of the graphs in supplementary fig. 1e

The authors mention in text about figure 1 that they are specifically analyzing p42 and that are multiple isoforms of C/EBP α . So, the authors should clarify which isoform they are overexpressing and why.

Results: “CEBPA expression was found reduced after 16 hours of TGF- β treatment and decreased expression was maintained during the 15 days (Fig S1e)”. Figure S1e does not have information about 16h.

Fig. 2a, indicate CEBPA TSS and direction. What does the double arrow around CEBPA represent?

Use more than one shRNA to confirm that experimental phenotypes are due to off-target effects (Fig. 2 – shSMAD3 and Fig. 5 – shCEBPA)

The authors report that C/EBP α protein expression is reduced by TGF- β after 24h. Can the authors provide quantification of the immunoblots on Fig. 1e and 2e, and provide a more representative blot if there is an average decrease across multiple experiments? Provide quantification of C/EBP α in Fig 1f.

In Fig. 3 how are C/EBP α and TGF- β regulated genes defined? Was there a threshold used in terms of expression levels? Also, define core EMT.

Provide more details about Fig. S2c – method, conditions, quantification. Define the assay.

In Fig. 4c, there is no expression of C/EBP α in EV cells. Provide different exposures.

Is data in Fig. S3b the same as Fig. 5a?

Figure S5c shows an immunoblot for 2D cultures of MCF10A cells after C/EBP α expression. However, it would be more informative to perform the immunoblot in 3D cultures. Joan S. Brugge's lab provides a protocol for this technique on its webpage.

Provide more details about the quantification performed on fig. 7d.

Fig. S6d is not cited in the text. What was the statistical test used, these data do not appear to have been analyzed in the same manner as other data in this manuscript.

Move text describing fig. S7a to the results section.

References: check author and journal in ref 27. Ref 3 and ref 45 are the same.

Organization of figure panels in supplemental figures: Does Fig. S2c refer to the 15-day experiment in Fig 4c? Move the panel to accompany related supplementary data. Does panel S3a refer to Fig 3? Consider moving to Fig S2.

In the discussion, the authors refer to "ectopic" and "constitutive expression", it is not clear to which experiments they are referring?

Reviewer #2 (Remarks to the Author):

This is a clear and compelling study showing for the first time the important role of C/EBP α reduction in the TGF β 1-induced EMT in HMLE human mammary cells. This is shown clearly in the HMLE cells, with supportive data in MCF10A, and finally in the PyMT organoid model, where the suppressive effects of C/EBP α on metastasis is shown without any influence on proliferation / tumor growth. The data appear very sound, although it would have been good to show whether EMT was also suppressed in the PyMT system by C/EBP α . Also, there should be evidence of reduced C/EBP α and C/EBP α -regulated genes in the abundant public data available around EMT.

1. There seems very little work on this TF and EMT, attesting to the novelty, however the following references are on topic and should be cited:

- PLoS One. 2016 Apr 6;11(4):e0153117. doi: 10.1371/journal.pone.0153117. eCollection 2016. C/EBP α Short-Activating RNA Suppresses Metastasis of Hepatocellular Carcinoma through Inhibiting EGFR/ β -Catenin Signaling Mediated EMT. Huan H1, Wen X1, Chen X2, Wu L1, Liu W3, Habib NA4, Bie P1, Xia F1.

- Cell Physiol Biochem. 2017 Jul 26;42(5):1779-1788. doi: 10.1159/000479457. [Epub ahead of print] C/EBP α Suppresses Lung Adenocarcinoma Cell Invasion and Migration by Inhibiting β -Catenin. Lu J1, Du C1, Yao J2, Wu B1, Duan Y1, Zhou L1, Xu D1, Zhou F1, Gu L1, Zhou H1, Sun Y1.

- Cell Physiol Biochem. 2015;35(2):816-28. doi: 10.1159/000369740. Epub 2015 Jan 30. Advanced oxidation protein products induce hypertrophy and epithelial-to-mesenchymal transition in human proximal tubular cells through induction of endoplasmic reticulum stress. Tang X1, Rong G, Bu Y, Zhang S, Zhang M, Zhang J, Liang X.

2. It is not clear from the materials and methods whether the various manipulations of the HMLE and MCF10A cells are clonal examples or pooled multiclonal populations?

3. In Fig 3e it is stated "For the selected genes, suppression of TGF- β -dependent targets was observed in the presence of C/EBP α expression. The heatmap in Fig 3d shows that the TGF β effects of some of the "core of cores" genes were not abrogated by forced expression of C/EBP α . These should be mentioned and discussed? Is this discordant with the RNA-Seq data? For e.g. IL1b was not apparently regulated by TGF β under EV, but was dramatically upregulated by TGF β in the C/EBP α -transfected cells. Others (e.g. S100A8) were not affected by TGF β in the EV cells, but were dramatically upregulated by C/EBP α , and still TGF β -repressed on top of that?

4. Was a partial morphological effect seen with shCEBP1_2? There is always some cancer regarding off-target effects? Even if only partial, they would be supportive? Or in the absence of that, could some data be provided using siRNAs smartpool as an alternative validation?

5. A similar concern could be raised on the sh_SMAD3 suppression with only 1 shRNA?

6. Is C/EBPa a core interactome gene? Was it shown to be regulated in the TGFb-treated HMLE cells in earlier published studies, such as Taube et al, Ref 25? Is it down-regulated in other EMTs shown in the literature?

7. Did the C/EBPa transduction of PyMT organoids affect their EMT status or propensity? As pointed out, they provide an opportunity to test the effects in another system, but EMT seems not to have been examined?

8. Why is VIM repressed with dox-induced C/EPBa plus TGFb1 vs TGFb1 alone (Fig 6)?

9. MET has been shown in some studies to assist colonisation at the secondary site, after EMT has helped metastatic cells to escape and survive. Are there any evidences to support a role for re-expression of C/EBPa in metastases?

10. A key issue will be consistency and clinical relevance. Is C/EBPa reduced in metastases compare to primary tumours? Is it reduced in more invasive / metastatic subgroups of breast cancer cell lines, EMT-enriched basal breast cancers?

Minor:

1. Page 3: the sentence "The majority of these EMT-effectors are responsible for transcriptional repression of CDH1 (E-cadherin) and induction of N-cadherin expression, of which contributes to weaker interactions and facilitates cell motility and invasion 9." Has awkward text "of which contributes"? maybe delete 'of'?

2. Page 3: cellular is listed after extracellular for ECM, delete 2nd cellular?
3. Page 5: "and 5 µg of shRNA contracts." Should be constructs?
4. Page 5: "To generate a conditionally regulated C/EBPα cells" delete "a"?
5. Page 6: "Rneasy Isolation Kit" should be "RNeasy"
6. Page 7: "Sodium dodecyl sulfate" should not be capitalised; Glycine should not be capitalised; "Sonication using covaris was" what is covaris ?
7. Page 8: "8 well glass bottom chamber (Ibidi) was coated with cultrex matrigel (R&D systems) for 1 hour at 37 degrees. How much Matrigel?; "MCF10A cells were platted" should be plated; "In the following day" should be "on" ; "Reduced growth factor basement membrane extract type 2," should not be capitalised;
8. Page 9: "using standard protocols and detailed above" "as detailed above"?;
9. Page 10; Is there a reference for "PO-block buffer "?; "Diaminobenzidine (DAB)" Diaminobenzidine (should not be capitalised?; "morphologically identify metastasis." Mestasases could better be plural?
10. Page 11: "tissueswere incubated" words are joined;
11. Page 12: "gatekeeper function, ," delete 2nd comma;
12. Page 13: Fig 2b, "SMAD3 binding to these regions was validated using chromatin-immunoprecipitation followed by qRT-PCR (ChIP-qRT-PCR)." Probably this is just PCR rather than RT-PCR?

13. Page 17: “This has been previously shown to spontaneously develop ductal mammary tumors that metastasize primarily to the lungs with resemblances of EMT-mediated tumorigenesis 4,27.” This sentence is not clear, following on from the preceding sentence (“we ectopically expressed C/EBP α and empty vector control in tumor organoids derived from YFP-positive PyMT breast tumors”). Presumably it refers to the use of organoids in vivo, however, as stated it refers to the ectopic expression, so this needs to be reworded.

14. What are the ‘EMT Core Genes’ (Fig 3/ Fig S2b) - are they the Core Interactome genes from Taube et al, Ref 25? I think so but it is not totally clear... Figure legend (Supp Fig 2) and text around Supp Fig 2b.

15. On Page 21, the following statement should list cited reference: “Induction of N-cadherin expression plays a pivotal role in EMT, and has been shown to be sufficient to promote cell migration and invasion in several cancer cell lines, regardless of E-cadherin expression.”

Reviewer #1 (Remarks to the Author):

In the manuscript “C/EBP α is (a) crucial determinant of epithelial homeostasis by preventing epithelial-to-mesenchymal transition”, the authors set out to identify transcription factors suppressed during the EMT process in human mammary cells. They find C/EBP α expression to be suppressed in response to TGF- β and show a negative relationship between expression of C/EBP α and several mesenchymal markers, including the transcription factor ZEB1, in two different mammary epithelial culture cell lines. Conversely, they find a positive relationship between expression of C/EBP α and the epithelial protein, E-cadherin. They go on to demonstrate that cells that have undergone a TGF- β -induced EMT reverted to an epithelial-like state upon C/EBP α re-expression under both 2D and 3D culture conditions. Finally, C/EBP α expression in the primary tumor prevented the formation of metastasis in an orthotopic mouse model. The authors conclude, “that C/EBP α is required for maintaining epithelial homeostasis” and “that C/EBP α is a master epithelial “gatekeeper” whose expression is required to prevent unwarranted mesenchymal transition”. The manuscript provides data supporting a negative relationship between levels of C/EBP α expression and induction of EMT, and supporting the idea that C/EBP α is able to suppress EMT. As I outline below, the manuscript in its current form requires significant revision, it lacks clearly stated rationale for several key experiments and provides insufficient data to support several key conclusions.

Firstly, we would like to apologize to the reviewer for the long delay in resubmitting our study. For the resubmission we wanted to properly address all the points raised by the reviewers, which were considerable. Unfortunately, several issues outside of our control resulted in us needing considerably more time than anticipated to complete this. However, we have now addressed the points raised by the reviewer and these are discussed in the point-by-point rebuttal below.

Major comments

1. In general the manuscript would greatly benefit from stronger rationale for the experiments conducted. Examples include: the selection of SMAD3 as a potential repressor of C/EBP α , is SMAD2 a poor candidate; the focus on C/EBP α instead of E2F2 or E2F8, C/EBP α itself has been shown to inhibit the cell cycle in myeloid cells; the use of PyMT model which has been shown to metastasize to the lung without undergoing EMT, Ref 4 (Fischer et al, Nature 2015).

As we demonstrate (**Figure S1f**), rapid repression of C/EBP α mRNA levels after 3 hours of TGF- β treatment suggests that TGF- β -mediated repression of C/EBP α is direct and exerted by its key transcriptional regulators, the SMADs. Phosphorylation-mediated activation of SMAD2 and SMAD3 occurs in response to the binding of TGF- β ligand. Unlike SMAD3, and due to an additional exon within its MH1 domain, SMAD2 fails to bind DNA and therefore lacks transcriptional activity, providing a rationale to explore the role of SMAD3 in the repression of C/EBP α expression. We have now added this discussion to the text (**page 15**).

We specifically choose to study C/EBP α since together with E2F2 it is the most down-regulated transcript after TGF- β -exposure (**Figure 1b**). E2F proteins are key players of the mammalian cell cycle machinery, while C/EBP α has been shown to play a role in regulating cell differentiation. Therefore, in the context of TGF- β -mediated EMT we felt it was more interesting to pursue and we have been able to indeed demonstrate a novel role for C/EBP α in epithelial homeostasis. This has now been mentioned on **page 14**.

The pathological contribution of EMT to metastasis development is currently under debate and although Fisher *et al.* ¹ have shown that Fsp1/vimentin-negative cells metastasize without undergoing EMT, the exclusive use of late mesenchymal markers precludes the assessment of the contribution of partial EMT to metastasis. Therefore, the use of these experiments is insufficient to conclude that the orthotopic injection of tumor cells into PyMT model fails to metastasize to the lung without undergoing EMT. Moreover, recent study has shown that Snail- or ZEB1-positive tumor cells derived from Snai1^{YFP/+}:MMTV PyMT animals do not express fsp1 nor vimentin ², indicating that fsp1 and vimentin are not universal markers of EMT. Additionally, we have previously shown that MMTV-PyMT animals develop metastases by the dissemination of cells that have undergone EMT without experimentally modifying EMT-regulators ³, supporting the use of MMTV-PyMT model as a breast cancer model to study the contribution of EMT to metastasis.

2. C/EBP α mRNA levels appear to decrease by ~75% after 24h of TGF- β treatment (Fig 1d), this decrease coincides with a dramatic increase in N-cadherin and fibronectin protein levels (Fig 1e). Unlike C/EBP α mRNA, the amount of C/EBP α protein appears unchanged in Fig 1d and mildly decreased in Fig 2e (provide quantification of C/EBP α protein levels relative to loading controls). How do the authors reconcile the dramatic increase in mesenchymal markers with a relatively modest decrease in C/EBP α ? Fig 1f shows a dramatic relocalization of C/EBP α out of the nucleus at 15 days, does this also occur at early time points? Investigation of how C/EBP α protein localization is regulated in response to TGF- β is likely to provide valuable insight. Does mutation of C/EBP α DNA binding or NLS disrupt its ability to suppress EMT?

We have quantified the C/EBP α protein levels. We have also performed a new time course to follow C/EBP α protein levels upon short incubations with TGF- β . This experiment indicates that C/EBP α protein remains relatively stable after short incubations (shorter than 24 hours) with TGF- β (**Rebuttal Figure 1a and 1b**). Moreover, immunostaining shows a gradual increase in cytoplasmic relocalization of C/EBP α after 8-72 hours incubation of TGF- β (**Rebuttal Figure 1b and 1c**). Thus, this suggests that in addition to a rapid transcriptional downregulation of C/EBP α by TGF- β , C/EBP α is also regulated at the post-translational level leading to a gradual cellular relocalization. We speculate that, in the first 24 hours multiple mechanisms determine the increase in mesenchymal markers, namely a decrease of C/EBP α mRNA levels, a cytoplasmic relocalization of C/EBP α protein and perhaps other post-translational mechanisms that affect C/EBP α transcriptional activity. Notably, we observed that C/EBP α is dynamically phosphorylated after short incubations with TGF- β as shown in **Rebuttal Figure 1d**. Our data support the conclusion that C/EBP α does affect EMT, but we feel it is beyond the scope of this manuscript to determine which post-transcriptional mechanisms are regulating C/EBP α activity.

Rebuttal Figure 1. (a) Results from immunoblotting analysis showing the effect of TGF- β (short) treatment on C/EBP α protein levels and respective quantification (non-significant). (b) Immunofluorescence staining showing the cellular localization of C/EBP α upon 0, 8, 24, 48 and 72 hours of TGF- β treatment in HMLE cells and (c) quantification of cytoplasmic signal normalized to the total area. d) Immunoblotting analysis of C/EBP α phosphorylation upon TGF- β (short) treatment using Phos-Tag gel technology. HMLE cells expressing empty vector (EV) or constitutive CEBPA were either left untreated or treated with 2.5ng/ml of TGF- β for 0, 2, 4, 6, 8 or 16 hours.

3. To explain the early decrease in C/EBP α mRNA, SMAD3 was assessed for binding to the C/EBP α locus in a role as putative repressor. As discussed above, the modest decrease in C/EBP α protein in response to TGF- β after 24h questions the relevance of inhibiting C/EBP α mRNA expression by SMAD3. To determine if SMAD3 is able to repress CEBPA expression, knock-down studies were done using a single hairpin. Could this observation be due to off target effects, demonstrate the result in 2d and 2e using multiple hairpins. Data shown in Fig 2d and 2e indicate an increase in basal levels of CEBPA expression

upon SMAD3 knock-down and a subsequent CEBPA decrease in response to TGF- β , both at the mRNA and protein levels. The decrease in Fig 2d is reported as not significant, however this appears to stem from an increase in experimental error. If the same level of standard deviation is achieved in the shSMAD3 group as shSCR would the reported conclusion stand? The basal increase in CEBPA may suggest a potential mechanism of long-term suppression, however the immediate decrease in CEBPA mRNA in response to TGF β suggest other regulation. Also considering C/EBP α protein stability at 24h, another mechanism of C/EBP α mediated N-cadherin and Fibronectin suppression requires consideration.

We have now confirmed this experiment using a second shRNA to knockdown SMAD3. This is provided in a revised **Figure 2d**. Using two independent shRNAs targeting SMAD3, we confirm that TGF- β -induced downregulation of C/EBP α is impaired, supporting that SMAD3 is required for its inhibition.

As discussed previously, C/EBP α activity might be regulated independently of its degradation. In addition to cellular relocalization, TGF- β treatment was able to induce a shift in the Phos-Tag electrophoretic mobility of C/EBP α (**Rebuttal Figure 1d**), suggesting that regulation of C/EBP α activity by phosphorylation may occur in the context of TGF- β -mediated EMT. Post-transcriptional regulation of C/EBP α tumor suppressive function has been previously described⁴. For instance, phosphorylation of residue Ser193 has been demonstrated to be critical for its tumor suppressive function in inhibiting cell cycle progression. Therefore, the induction of cytoplasmic sequestration and potential effect of post-transcriptional modifications on C/EBP α might alleviate C/EBP α -mediated repression of mesenchymal genes such as CDH2. We have now discussed this briefly in the discussion on **page 23**.

4. ChIP-seq data indicated two regions where SMAD3 was enriched in proximity to the CEBPA locus. Enrichment was validated using ChIP-qPCR. To improve robustness of the validation experiment, consider the addition of a TGFBR1/ALK5 inhibitor (eg SB431542) to limit basal SMAD3 activation/nuclear accumulation in the control. Do the two regions identified contain SMAD3 binding sequence motifs? Is SMAD3 binding these regions directly?

We would respectfully argue that the combination of ChIP and ChIP qRT-PCR is both unbiased and robust and thus do not believe that addition of a TGFBR1 inhibitor would increase the confidence in these experiments. We have provided a new **Figure 2a**, which shows that both regions do indeed contain SMAD3 primary binding sequence motif supporting that SMAD3 binds these regions directly.

5. TGF- β is used to induce EMT throughout the manuscript. Does overexpression of C/EBP α (as shown in Fig 3e) affect SMAD2/3 phosphorylation at 1 and 24 hrs after TGF- β treatment? Could the decrease in TGF- β response reported in Fig 3e be due to interference with the cell's ability to signal properly?

To determine whether C/EBP α affects TGF- β -mediated SMAD3 phosphorylation we performed immunoblotting analysis, which are shown in **Supplementary Figure 3c** and discussed on **page 17**. These experiments indicate that C/EBP α overexpression does not interfere with general TGF- β signaling as SMAD3 phosphorylation seems largely unaltered.

What was the period of time between sorting of GFP positive cells and initiation of the experiment?

Cells were expanded for 2 weeks after sorting before experiments, this has now been added to the Materials and Methods.

Overexpression of C/EBP α appears to affect basal levels of several mesenchymal associated mRNAs. What are the relative levels of C/EBP α overexpression to endogenous protein/mRNA in Fig 3?

As visualized in **Figures S3b** and **S6a**, ectopic C/EBP α is expressed around 200 times more compared to endogenous levels.

6. The authors state that in cancer, EMT is important for the acquisition of migratory and invasive properties and show that genes common to C/EBP α and TGF- β expression profiles are related to ECM remodeling and cell migration. Does overexpression/knock-down of C/EBP α affect migration and invasion of TGF- β stimulated epithelial cells in Boyden chamber assays?

In order to answer this question, we have performed wound-healing migration assays, which are now included in **Figure 4d**. These experiments show that C/EBP α overexpression impairs TGF- β -mediated migration on different extracellular matrices.

7. The manuscript states that C/EBP α : 1) is crucial to maintain epithelial architecture and 2) prevent EMT. The authors reach these conclusions without providing a mechanism showing how C/EBP α promotes expression of epithelial genes or limits expression of mesenchymal genes. A potential mechanism of ZEB1 transcriptional inhibition is highlighted in the results and discussion, however no data is provided to support such a mechanism. Evidence of such regulation in epithelial cells would be of great interest. Recent work in adipose cells has shown that ZEB1 is a positive regulator of C/EBP α and adipocyte differentiation (Gubelmann et al, Elife 2015), notably knockdown of ZEB1 greatly decreased C/EBP α in these cells. One might expect that C/EBP α must be regulated differently in epithelial cells undergoing EMT, where C/EBP α is decreased, as ZEB1 is increased. Can the authors comment on the regulatory divergence among cell types?

Since C/EBP α is a transcription factor it may act by directly affecting epithelial genes or by affecting other factors that maintain epithelial genes and repress expression of mesenchymal genes. Indeed, we have shown using luciferase reporter assays that C/EBP α can bind and activate CDH1 (E-cadherin) promoter (**Figure S8e**). In addition, as mentioned C/EBP α and ZEB1 are inversely related in mammary epithelial cells, such that C/EBP α could affect EMT genes by regulation of ZEB1.

With regard to the opposite regulation of C/EBP α and ZEB1 in adipocytes we can only speculate that different co-factors mediate the effects of ZEB1 on C/EBP α in different cell types. However, it must be said that it has to be determined whether ZEB1 affects C/EBP α levels in mammary epithelial cells and if so, whether it upregulates or downregulates C/EBP α in these cells. It is not inconceivable that in mammary epithelial cells ZEB1 can still positively regulate C/EBP α mRNA levels in a negative-feedback loop that may aid mesenchymal cells during developmental processes to reacquire epithelial traits.

8. On page 16, the authors state, "C/EBP α is indispensable for epithelial cells to retain their epithelial traits". However, the sole epithelial marker analyzed, E-cadherin (in panel 5c) showed a ~20% decrease, while mesenchymal markers showed more pronounced

changes. Is C/EBP α playing a direct role in maintaining expression of epithelial genes or is it more indirect, by preventing expression of mesenchymal transcription factors and genes, which in turn suppress epithelial traits?

We have now included analyses of to show that C/EBP α also regulates expression of additional epithelial markers in **Figure S3b** and **Figure S6a**. Importantly, the depletion of C/EBP α in both HMLE and MCF10A epithelial cells was sufficient to induce significant morphological and structural alterations towards a mesenchymal-like phenotype (**Figure 5b** and **Figure S4c**), supporting that C/EBP α is indeed important for epithelial cells to retain their epithelial traits. The role of C/EBP α in maintaining epithelial homeostasis is likely exerted by the combination of direct and indirect regulation of epithelial genes and transcription factors. We have observed that C/EBP α is able to induce CDH1 reporter activity using a dual luciferase reporter assay (**Figure S8e**). Moreover, by inhibiting the transcription of mesenchymal transcription factors, such as Zeb1, C/EBP α can alleviate E-cadherin repression and contribute to the restoration of E-cadherin expression. We have further updated the following section in the discussion:

“Maintenance of epithelial traits by C/EBP α can be envisioned by its direct involvement in the transcriptional activation of CDH1 (E-cadherin) and repression of ZEB1 (Fig. S8e and Fig. 8a) and/or by the activation of an intermediate player responsible for transcriptionally repressing pro-oncogenic genes such as CDH2 and MMP2 (Fig. 8b). Additionally, we cannot exclude that C/EBP α itself cannot directly bind to the regulatory regions of these genes promoting its repression, as C/EBP transcription factors are both activators and repressors ⁵ (Fig. 8c).“

9. Cells from PyMT tumors must extravasate, survive and intravasate into the lung tissue prior to formation of the metastases observed in Fig 7. Was blood drawn to determine if EV and C/EBP α tumors released equal number of invasive cells into the blood stream? Perhaps C/EBP α overexpression allows fewer cells to escape the primary tumor, a possibility considering the gene expression profile associated with C/EBP α . Is it possible to analyze the epithelial/mesenchymal status of these cells? This is important to address.

We have performed this experiment (**Rebuttal Figure 2**). Due to the low number of circulating tumor cells (CTCs) recovered, for statistical significance a very large number of animals would be required. Due to restricted animal regulations in the Netherlands, this has been problematic. Although not statistically significant, we observed a reduction in the number of CTCs in mice injected with C/EBP α -overexpressing organoids, suggesting that C/EBP α suppresses cell intravasation. Accordingly, C/EBP α expression results in the inhibition of important components involved in this process, including CDH2 (N-cadherin), FN1 (fibronectin) and PLAU (urokinase or uPA) (**Figures 4b-c** and RNA-sequencing data, respectively).

Rebuttal Figure 2 – Assessment of the number of circulating tumor cells (CTCs) present in 900 μ L of blood obtained from mice injected with control or C/EBP α -overexpressing tumor organoids.

Minor comments and suggestions:

Provide table with the sequence of primers used in this study.

Table is provided on page 7.

Title: ‘epithelial homeostasis’ evokes the role of C/EBP α in maintenance of epithelial tissues, which is not the focus of this paper. The manuscript focuses on its role in limiting EMT aided metastasis instead, consider revising.

We appreciate the reviewer’s comment and have now replaced the word “homeostasis” with “maintenance” in the title.

Introduction: page 3, line 3rd from the end – “establishing” instead of “stablishing”

This has been changed in the revised manuscript.

Results: First line on page 12, extra comma. First sentence on page 14, editing, delineate or define. Last line on page 14, last line – “difference” instead of “differences”

This has been changed in the revised manuscript.

In the material and methods, indicate the concentration of cell media components.

Information has been added to the Material and Methods.

Figure 1c is more appropriate for the supplement.

Figure 1c has been placed in the Supplementary Figures as Figure S1b as requested by the reviewer.

In supplementary fig. 1, figure 5 and 6 note the use of “,” (commas) instead of “.” (decimals).

This has been changed in the revised manuscript.

There are formatting problems in the x axis of the graphs in supplementary fig. 1e

This has been changed in the revised manuscript.

The authors mention in text about figure 1 that they are specifically analyzing p42 and that are multiple isoforms of C/EBP α . So, the authors should clarify which isoform they are overexpressing and why.

This information has been added to the revised manuscript.

Results: “CEBPA expression was found reduced after 16 hours of TGF- β treatment and decreased expression was maintained during the 15 days (Fig S1e)”. Figure S1e does not have information about 16h.

Axis labeling error has been rectified (0.5 should be 0.7 days corresponding to 16 hours).

Fig. 2a, indicate CEBPA TSS and direction. What does the double arrow around CEBPA represent?

CEBPA TSS and respective direction have been added to revised Figure 2.

Use more than one shRNA to confirm that experimental phenotypes are due to off-target effects (Fig. 2 – shSMAD3 and Fig. 5 – shCEBPA)

Experiments with a second shSMAD3 have been performed and included as revised Fig. 2d. siRNA pool targeting CEBPA was used to confirm experimental phenotypes obtained from shRNA CEBPA and included as Fig. S4a.

The authors report that C/EBP α protein expression is reduced by TGF- β after 24h. Can the authors provide quantification of the immunoblots on Fig. 1e and 2e, and provide a more representative blot if there is an average decrease across multiple experiments? Provide quantification of C/EBP α in Fig 1f.

Quantification of C/EBP α expression is now provided in revised figures.

In Fig. 3 how are C/EBP α and TGF- β regulated genes defined? Was there a threshold used in terms of expression levels? Also, define core EMT.

C/EBP α and TGF- β regulated genes were defined by the genes that are differentially regulated by TGF- β in the control cells and are affected (down- and upregulated) by C/EBP α overexpression in TGF- β treatment conditions excluding the genes that were affected by C/EBP α overexpression in untreated conditions. A fold change of value ≥ 1.5 ($\log_2FC \geq 0.58$) and ≤ 0.72 ($\log_2FC \leq -0.48$) were used as thresholds for this analysis. This information has been added to the revised manuscript.

The core EMT genes were obtained from Taube et al. (2010) where they identified an EMT core signature derived from changes in gene expression shared by up-regulation of Gsc, Snail, Twist, and TGF- β 1 and by down-regulation of E-cadherin.

Provide more details about Fig. S2c – method, conditions, quantification. Define the assay. In Fig. 4c, there is no expression of C/EBP α in EV cells. Provide different exposures.

In Fig. S2c, equal numbers of control or C/EBP α -overexpressing HMLE cells were seeded on a tissue-culture treated 6 well plate. After 4 days, cell dissociation using trypsin was performed and the number of cells in both experimental groups was analyzed using trypan blue stain. This information has been added to the Material & Methods.

Based on the qRT-PCR analysis, C/EBP α mRNA levels in C/EBP α expression system is around 200 times higher than in control cells suggesting that similar ratio is observed at the protein level. Therefore, is not possible to present a blot displaying C/EBP α expression levels in EV without complete over-exposure.

Is data in Fig. S3b the same as Fig. 5a?

Yes, Fig. S3b has been removed from the revised manuscript.

Figure S5c shows an immunoblot for 2D cultures of MCF10A cells after C/EBP α expression. However, it would more informative to perform the immunoblot in 3D cultures. Joan S. Brugge's lab provides a protocol for this technique on its webpage.

For immunoblotting analysis, we would need to drastically scale up the amount of 8-well chamber slide wells per condition in order to get sufficient material to analyze all the proteins of interest. This requires a serious amount of matrigel and costs that we believe that are unnecessary as we provide the characterization of these cells in Fig. S6c.

Provide more details about the quantification performed on fig. 7d.

Sections of primary tumor of mice injected with C/EBP α -expressing tumor organoids and sections of lungs of mice injected with control or C/EBP α -expressing tumor organoids were analyzed for C/EBP α and YFP expression by immunofluorescence. The quantified C/EBP α expression levels in YFP-positive primary tumor cells was used as a set-point (100%) to assess the expression levels of C/EBP α in YFP-positive metastatic cells from both groups.

Fig. S6d is not cited in the text. What was the statistical test used, these data do not appear to have been analyzed in the same manner as other data in this manuscript.

Fig. S6d is now cited on the revised manuscript on page 20. T-Test was used as statistical test for this figure, as for any other figure in this manuscript where two groups are compared. This information is available in the Material and Methods on page 11.

Move text describing Fig. S7a to the results section.

Fig. S7a has been incorporated in Fig. S3b.

References: check author and journal in ref 27. Ref 3 and ref 45 are the same.

These have been rectified in the revised manuscript.

Organization of figure panels in supplemental figures: Does Fig. S2c refer to the 15-day experiment in Fig 4c? Move the panel to accompany related supplementary data. Does panel S3a refer to Fig 3? Consider moving to Fig S2.

Fig. S2c refers to the analysis of cell growth between control or C/EBP α -expressing HMLE cells in untreated conditions and is now Fig.S3a. Figure S3a refers to Figure 4.

In the discussion, the authors refer to “ectopic” and “constitutive expression”, it is not clear to which experiments they are referring?

We used both terms to refer to C/EBP α overexpression. In order to avoid confusion, we have consistently used “constitutive expression” in the discussion of the revised manuscript.

Reviewer #2 (Remarks to the Author):

Firstly, we would like to apologize to the reviewer for the long delay in resubmitting our study. For the resubmission we wanted to properly address all the points raised by the reviewers, which were considerable. Unfortunately, several issues outside of our control resulted in us needing considerably more time than anticipated to complete this. However, we have now addressed the points raised by the reviewer and these are discussed in the point-by-point rebuttal below.

This is a clear and compelling study showing for the first time the important role of C/EBP α reduction in the TGFB1-induced EMT in HMLE human mammary cells. This is shown clearly in the HMLE cells, with supportive data in MCF10A, and finally in the PyMT organoid model, where the suppressive effects of C/EBP α on metastasis is shown without any influence on proliferation / tumor growth. The data appear very sound, although it would have been good to show whether EMT was also suppressed in the PyMT system by C/EBP α . Also, there should be evidence of reduced C/EBP α and C/EBP α -regulated genes in the abundant public data available around EMT.

We have now included this data on page 14 and 21 of the revised manuscript.

1. There seems very little work on this TF and EMT, attesting to the novelty, however the following references are on topic and should be cited:

• PLoS One. 2016 Apr 6;11(4):e0153117. doi: 10.1371/journal.pone.0153117. eCollection 2016. C/EBP α Short-Activating RNA Suppresses Metastasis of Hepatocellular Carcinoma through Inhibiting EGFR/ β -Catenin Signaling Mediated EMT. Huan H1, Wen X1, Chen X2, Wu L1, Liu W3, Habib NA4, Bie P1, Xia F1.

• Cell Physiol Biochem. 2017 Jul 26;42(5):1779-1788. doi: 10.1159/000479457. [Epub ahead of print] C/EBP α Suppresses Lung Adenocarcinoma Cell Invasion and Migration by Inhibiting β -Catenin. Lu J1, Du C1, Yao J2, Wu B1, Duan Y1, Zhou L1, Xu D1, Zhou F1, Gu L1, Zhou H1, Sun Y1.

• Cell Physiol Biochem. 2015;35(2):816-28. doi: 10.1159/000369740. Epub 2015 Jan 30. Advanced oxidation protein products induce hypertrophy and epithelial-to-mesenchymal transition in human proximal tubular cells through induction of endoplasmic reticulum stress. Tang X1, Rong G, Bu Y, Zhang S, Zhang M, Zhang J, Liang X.

We have cited and discussed these references in the revised manuscript.

2. It is not clear from the materials and methods whether the various manipulations of the HMLE and MCF10A cells are clonal examples or pooled multiclonal populations?

All experiments in this manuscript have been carried out with pooled multiclonal populations. We have now also made this clear in the materials and methods.

3. In Fig 3e it is stated "For the selected genes, suppression of TGF- β -dependent targets was observed in the presence of C/EBP α expression. The heatmap in Fig 3d shows that the TGFB effects of some of the "core of cores" genes were not abrogated by forced

expression of C/EBP α . These should be mentioned and discussed? Is this discordant with the RNA-Seq data? For e.g. IL1b was not apparently regulated by TGFB under EV, but was dramatically upregulated by TGFB in the C/EBP α -transfected cells. Others (e.g. S100A8) were not affected by TGFB in the EV cells, but were dramatically upregulated by C/EBP α , and still TGFB-repressed on top of that?

We performed Go-enrichment analysis for the 20 genes included in Fig. 3d that were upregulated in C/EBP α -overexpressing cells (see Rebuttal Figure 3). Pathways involved in leukocyte chemotaxis and leukocyte degranulation were found enriched. C/EBP α is known to play a pivotal role in immune cell fate and differentiation as well as in the regulation of genes involved in inflammation, including IL1 β ⁶. Therefore, some of the listed genes are known C/EBP α targets explaining the cumulative increase of expression on C/EBP α -overexpressing cells treated with TGF- β . On the other hand, genes like S100A8 and SERPINB2 have been shown to be regulated by TGF- β , therefore we cannot exclude that the lack of responsiveness in EV cells are due to technical effects rather than physiological^{7,8}. We have included the following discussion on page 14/15:

Closer examination at these 37 genes shown two main clusters: genes that are upregulated by TGF- β in control cells but repressed in the presence of C/EBP α overexpression and genes which expression is increased in C/EBP α -overexpressing cells in both untreated and TGF- β treated conditions. GO-Term enrichment analysis of these two clusters revealed that first cluster is enriched for processes involved in cell-matrix adhesion and organization whereas second cluster is related to processes involved in inflammatory response (data not shown). We focused on genes comprising the first cluster as initial steps of EMT-induced tumor progression requires extensive cell-cell and cell-matrix re-organization.

Rebuttal Figure 3 – GO-Term enrichment analysis of the 20 genes comprising cluster 2 and which expression is increased in C/EBP α -overexpressing cells in both untreated and TGF- β treated conditions. Metascape software was used for this analysis.

4. Was a partial morphological effect seen with shCEBP1_2? There is always some cancer regarding off-target effects? Even if only partial, they would be supportive? Or in the absence of that, could some data be provided using siRNAs smartpool as an alternative validation?

To ascertain that the effects seen with the shRNA are not due to off-target effects we have used an siRNA approach. Indeed, upon transfection of HMLEs with an siRNA smartpool targeting CEBCPA we confirmed the alteration in EMT markers previously found with the shRNA. This experiment is now presented as Fig. S4a.

5. A similar concern could be raised on the sh_SMAD3 suppression with only 1 shRNA?

We have now used a second shRNA to knockdown SMAD3 that confirmed our previous findings. This experiment is provided in the revised Fig. 2d.

6. Is C/EBP α a core interactome gene? Was it shown to be regulated in the TGF β -treated HMLE cells in earlier published studies, such as Taube et al, Ref 25? Is it down-regulated in other EMTs shown in the literature?

C/EBP α is not listed in the EMT-interactome signature described by Taube *et al.* C/EBP α would only be considered as a core EMT-interactome gene if its expression would be downregulated in HMLEs overexpressing Gsc, Snail, Twist, TGF- β 1 and depleted for E-cadherin altogether (3 replicates per condition). However, upon analysis of the microarray data obtained from Taube *et al.*, we observed that C/EBP α is downregulated in nearly all conditions that induce EMT (see **Rebuttal Figure 4**) including forced expression of TGF- β 1. Presumably CEBP α is not part of the EMT-interactome signature due to the expression levels in the replicate GSM595329. Nonetheless, these data support the concept that EMT induces a downregulation of C/EBP α in mammary epithelial cells. CEBPA expression data in control and TGF- β 1-overexpressing HMLE cells has been included in the revised manuscript as Fig. S1c.

Rebuttal Figure 4 – Analysis of CEBPA expression from public microarray data available from Taube et al. (Accession number GSE24202).

7. Did the C/EBP α transduction of PyMT organoids affect their EMT status or propensity? As pointed out, they provide an opportunity to test the effects in another system, but EMT seems not to have been examined?

We have also analyzed the percentage of E-cadherin^{low} and E-cadherin^{high} cells present in control tumor organoids or tumor organoids overexpressing C/EBP α . Similar to the results observed from the primary tumors, C/EBP α -expressing tumor organoids display lower percentage of E-cadherin^{low} cells compared to control group (see **Rebuttal Figure 5**). Once again, the majority of tumor cells comprising the tumor organoids are E-cadherin^{high} making the detection of changes in E-cadherin^{low} cell numbers challenging.

Rebuttal Figure 5 – Control or C/EBP α -overexpressing tumor organoids were dissociated to single cells, stained for extracellular E-cadherin (Alexa 647) and analyzed for YFP, CFP and E-cadherin expression by flow cytometry analyses. YFP-positive and DAPI-negative cells were further analyzed for low and high E-cadherin expression using both CFP and extracellular E-cadherin expression.

8. Why is VIM repressed with dox-induced C/EBP α plus TGF β 1 vs TGF β 1 alone (Fig 6)?

In most of our analyses, C/EBP α overexpression does not affect vimentin levels. In this experimental setting, we can only speculate that there is a synergetic effect of doxycycline and C/EBP α in inducing vimentin repression. Doxycycline has been reported to affect the expression of certain genes, including vimentin⁹. This suggests that, C/EBP α is a weak repressor of vimentin in physiological conditions and requires other factors to repress vimentin.

9. MET has been shown in some studies to assist colonisation at the secondary site, after EMT has helped metastatic cells to escape and survive. Are there any evidences to support a role for re-expression of C/EBP α in metastases?

No evidence has so far been published regarding this matter. We observed that introduction of C/EBP α in TGF- β -induced mesenchymal cells leads to partial MET (Fig. 6).

10. A key issue will be consistency and clinical relevance. Is C/EBP α reduced in metastases compare to primary tumours? Is it reduced in more invasive / metastatic subgroups of breast cancer cell lines, EMT-enriched basal breast cancers?

Indeed, it is interesting to assess the clinical relevance of the role of C/EBP α in breast cancer. Therefore, we investigated CEBPA expression in the public available database HCMDDB (Human Cancer Metastasis Database) and several human breast cancer cell lines. The following was added to the discussion on page 26:

“Importantly, our observations are supported by clinical data of breast cancer patients where CEBPA expression is found higher in primary normal tissue compared to primary tumor and markedly lower in primary tumors with metastasis compared to primary tumor without metastasis (Fig. S8a and Fig. S8b). These data support the notion that C/EBP α is an inhibitory factor to tumor development and metastasis formation in a clinical setting. Moreover, we have also evaluated C/EBP α expression levels in various breast cancer cell lines and this also showed lower C/EBP α expression in metastatic breast tumor cell lines, such as MDA-MB-231 and MDA-MB-436, in comparison to untransformed HMLE cells (Fig. S8c and Fig. S8d).”

Minor:

1. Page 3: the sentence “The majority of these EMT-effectors are responsible for transcriptional repression of CDH1 (E-cadherin) and induction of N-cadherin expression, of which contributes to weaker interactions and facilitates cell motility and invasion 9.” Has awkward text “of which contributes”? maybe delete ‘of’?

This has been changed in the revised manuscript.

2. Page 3: cellular is listed after extracellular for ECM, delete 2nd cellular?

This has been changed in the revised manuscript.

3. Page 5: “and 5 μ g of shRNA contracts.” Should be constructs?

This has been changed in the revised manuscript.

4. Page 5: “To generate a conditionally regulated C/EBP α cells” delete “a”?

This has been changed in the revised manuscript.

5. Page 6: “Rneasy Isolation Kit” should be “RNeasy”

This has been changed in the revised manuscript.

6. Page 7: “Sodium dodecyl sulfate” should not be capitalised; Glycine should not be capitalised; “Sonication using covaris was” what is covaris ?

This has been changed in the revised manuscript.

Covaris is an ultrasonicator. This information has been added to the manuscript.

7. Page 8: “8 well glass bottom chamber (Ibidi) was coated with cultrex matrigel (R&D systems) for 1 hour at 37 degrees. How much Matrigel?; “MCF10A cells were plated” should be plated; “In the following day“ should be “on” ; “Reduced growth factor basement membrane extract type 2,” should not be capitalized.

This has been changed in the revised manuscript.

8. Page 9: “using standard protocols and detailed above” “as detailed above”?

This has been changed in the revised manuscript.

9. Page 10; Is there a reference for “PO-block buffer “?; “Diaminobenzidine (DAB)” Diaminobenzidine (should not be capitalised?; “morphologically identify metastasis.” Mestasases could better be plural?

This has been changed in the revised manuscript.

10. Page 11: “tissueswere incubated” words are joined.

This has been changed in the revised manuscript.

11. Page 12: “gatekeeper function, ,” delete 2nd comma.

This has been changed in the revised manuscript.

12. Page 13: Fig 2b, “SMAD3 binding to these regions was validated using chromatin-immunoprecipitation followed by qRT-PCR (ChIP-qRT-PCR).” Probably this is just PCR rather than RT-PCR?

We indeed performed qRT-PCR.

13. Page 17: “This has been previously shown to spontaneously develop ductal mammary tumors that metastasize primarily to the lungs with resemblances of EMT-mediated tumorigenesis 4,27.” This sentence is not clear, following on from the preceding sentence (“we ectopically expressed C/EBP α and empty vector control in tumor organoids derived from YFP-positive PyMT breast tumors“). Presumably it refers to the use of organoids in vivo, however, as stated it refers to the ectopic expression, so this needs to be reworded.

Wording regarding this section has been revised.

14. What are the ‘EMT Core Genes’ (Fig 3/ Fig S2b) - are they the Core Interactome genes from Taube et al, Ref 25? I think so but it is not totally clear... Figure legend (Supp Fig 2) and text around Supp Fig 2b.

The EMT Core Genes are indeed the Core Interactome genes from Taube et al, Ref 25. Wording regarding this point has been revised.

15. On Page 21, the following statement should list cited reference: “Induction of N-cadherin expression plays a pivotal role in EMT, and has been shown to be sufficient to promote cell migration and invasion in several cancer cell lines, regardless of E-cadherin expression.”

References have been listed in the revised manuscript.

References:

1. Fischer, K. R. *et al.* Epithelial-to-mesenchymal transition is not required for lung metastasis but contributes to chemoresistance. *Nature* **527**, 472–476 (2015).
2. Arising, K. R. & Fischer. Upholding a role for EMT in breast cancer metastasis. *Nat. Publ. Gr.* **547**, 472–476 (2017).
3. Beerling, E. *et al.* Plasticity between Epithelial and Mesenchymal States Unlinks EMT from Metastasis-Enhancing Stem Cell Capacity. *Cell Rep.* **14**, 2281–2288 (2016).
4. Lourenço, A. R. & Coffey, P. J. A tumor suppressor role for C/EBP α in solid tumors: more than fat and blood. *Oncogene* (2017). doi:10.1038/onc.2017.151
5. Lekstrom-Himes, J. & Xanthopoulos, K. G. Biological role of the CCAAT/enhancer-binding protein family of transcription factors. *J. Biol. Chem.* **273**, 28545–8 (1998).
6. Poli, V. The role of C/EBP isoforms in the control of inflammatory and native immunity functions. *J. Biol. Chem.* **273**, 29279–82 (1998).
7. Rahimi, F., Hsu, K., Endoh, Y. & Geczy, C. L. FGF-2, IL-1 β and TGF- β regulate fibroblast expression of S100A8. *FEBS J.* **272**, 2811–2827 (2005).
8. Elsafadi, M. *et al.* SERPINB2 is a novel TGF β -responsive lineage fate determinant of human bone marrow stromal cells. *Sci. Rep.* **7**, 10797 (2017).
9. Qin, Y. *et al.* Doxycycline reverses epithelial-to-mesenchymal transition and suppresses the proliferation and metastasis of lung cancer cells. *Oncotarget* **6**, 40667–79 (2015).

Reviewers' comments:

Reviewer #1 (Remarks to the Author):

The authors have now addressed most of my major concerns and all of my minor concerns. I have two remaining requests. In point 2, I agree with the authors that investigating post-translational modifications is beyond the scope of this report, however, the disconnect that has been noted in point #2 should be addressed at some level and cytoplasmic-nuclear redistribution would be one way to account for this lack of correlation. Confocal imaging and quantitation, plus and minus C/EBP α knockdown would support the authors' conclusions. This data should be provided for the readers. Also, in point 6, the authors do not address invasive capacity with wound healing assays. At a minimum, a Boyden chamber cell invasion assay (through matrix like matrigel) would address this question.

Reviewer #2 (Remarks to the Author):

I continue to feel that this is a clear and compelling study showing for the first time the important role of C/EBP α reduction in the TGF β 1-induced EMT in HMLE human mammary cells. This is shown clearly in the HMLE cells, with supportive data in MCF10A, and finally in the PyMT organoid model, where the suppressive effects of C/EBP α on metastasis is shown without any influence on proliferation / tumor growth. The authors have addressed a number of additional major points raised in terms of literature supporting the important role of C/EBP α in modulating TGF β -1 – induced EMT, additional analyses / details, use of additional shRNA or siRNA Smartpools as an alternative validation of sh-RNA data, and most importantly, supporting evidence from clinical breast cancer metastasis data and breast cancer cell line systems. The minor issues raised have also been addressed.

The manuscript is much improved / refined with the Responses to both reviewers, and I am happy with the corrections made to Reviewer 2. I have a number of minor points to make below:

Minor:

1. 2nd, 3rd, 4th authors contributed equally, but not 1st-?OK
2. Mention name of 'well established mouse model of emt-driven metastasis' in the abstract
3. The transcription factors, known as EMT-effectors: There is a push for these to be called EMT-TFs; can the authors pre-empt this and adjust?
4. If C/EBP α levels are reduced in the majority of breast cancer specimens (17), why so little EMT in BrCa generally?
5. Response to this optional. C/EBP α expression levels (mRNA) are not much reduced in human breast cancer cell lines in the Basal or Mesenchymal subgroups compared to Luminal subgroup (Daemon et al, 2003, PMID 24176112). This is, however, more change than seen in CEBP-beta, delta or gamma. It may be worth a comment. The modest (but apparent) reduction of CEBPA mRNA across these subgroups may support the proposed post-Transcriptional regulation mentioned in the Discussion. See also Figs S8c,d...
6. In the newly added sentence "Cells were expanded for 2 weeks after sorting before experiments. The expression of C/EBP α p42 isoform was confirmed by Immunoblotting analysis." I don't think Immunoblotting should be capitalized?

7. shRNA targeting CEBPA (Sigma-Aldrich, TRCN000007304)... Thank-you for adding the siRNA smartpool date now. Later it states: "CEBPA-depleted HMLE cells were generated using two independent shRNA constructs" but that "expression showed strong reduction in CEBPA mRNA levels in only one of the shRNAs used". The catalog number of the one that didn't work so well should be included.

8. Figure 5a; the statistical significance associated with * and ** is not defined? Please check all Figures for this.

9. In the new text "we choose to further explore the role of C/EBP α as it has been shown to play a role in cell differentiation" I would suggest to add 'especially' at the start of the new text?

10. Under "Wound healing migration assays

24-well plates were coated with Collagen I, Collagen IV or Fibronectin" I think the proteins should not be capitalised?

11. I think the following new statement "Mammary tumors from this model, and similar to human ductal carcinomas, consist of a very small percentage of E-cadherinlow cells which makes detecting changes in Ecadherinlow cell number challenging" could be references, e.g. ref 4? Fig 7B results should be described differently – If not significant it cannot be called lower? The most important result still holds - The reduced metastasis seen.

12. In the Discussion, the following sentence "In accordance with these observations, constitutive expression of C/EBP α during TGF- β -mediated EMT contributed to repression of ZEB1 (Fig. S3b), N-cadherin and MMP2 as well as maintenance of E-cadherin levels, however no significant repression of vimentin was detected." has a Figure reference for Zeb1 but not the other factors (Fig 4B?)

13. In the Discussion, the following statement is made: "Moreover, by using an established mice model of EMT-induced breast cancer, we validated the tumor suppressive role of C/EBP α in vivo." However, Figure 7A clearly shows no effect on Primary tumor growth? Should this sentence read 'metastasis' rather than 'tumor'? Metastasis is clearly reduced (Figure 7c, 7d)...

14. Similarly, just further in the Discussion, it is stated 'These data support the notion that C/EBP α is an inhibitory factor for tumor development and metastasis formation in a clinical setting. Should 'tumor development' be included here.

15. Figure 8 indicates a cell exhibiting partial-EMT, with reduced C/EBP α ; what is the scenario anticipated in the full EMT shown in the adjacent cell?

Reviewers' comments:

Reviewer #1 (Remarks to the Author):

The authors have now addressed most of my major concerns and all of my minor concerns. I have two remaining requests. In point 2, I agree with the authors that investigating post-translational modifications is beyond the scope of this report, however, the disconnect that has been noted in point #2 should be address at some level and cytoplasmic-nuclear redistribution would be one way to account for this lack of correlation. Confocal imaging and quantitation, plus and minus C/EBP α knockdown would support the authors conclusions. This data should be provided for the readers.

As requested by the reviewer, we have now quantified C/EBP α nuclear and cytoplasmic levels from our confocal images (included in **Figure 1e** and **Figures S1h-i**). We have detailed and discussed these results in the revised manuscript (pages 6, 14 and 15). These data demonstrate that there is some cytoplasmic relocation of C/EBP α after TGF β -treatment that may to some degree explain the discrepancy between C/EBP α protein levels and transcriptional responses. As we highlighted in our previous rebuttal there are likely multiple mechanisms underlying these observations. We feel that quantification of nuclear and cytoplasmic C/EBP α levels under of conditions of C/EBP α knockdown would not be informative as knockdown results in inhibition of total C/EBP α expression.

Also, in point 6, the authors do not address invasive capacity with wound healing assays. At a minimum, a Boyden chamber cell invasion assay (through matrix like matrigel) would address this question.

While we agree with the reviewer that we do not specifically address invasive capacity with wound healing assays, this was not part of the message of our study. Our approach was based on RNA-sequencing data where we observed that genes associated with cell migration were regulated by C/EBP α in the context of TGF- β . Wound healing assays using different cellular matrices are a good approach to assess the impact of C/EBP α on cell migration (**Figure 4d**). Furthermore, we have now also included analysis of the impact of C/EBP α knockdown on cell migration using transwell assays to further validate our results. In line with our previous observations, C/EBP α knockdown was sufficient to increase cell migration of MCF10A cells (included as **Figure S4f**).

Reviewer #2 (Remarks to the Author):

I continue to feel that this is a clear and compelling study showing for the first time the important role of C/EBP α reduction in the TGFB1-induced EMT in HMLE human mammary cells. This is shown clearly in the HMLE cells, with supportive data in MCF10A, and finally in the PyMT organoid model, where the suppressive effects of C/EBP α on metastasis is shown without any influence on proliferation / tumor growth. The authors have addressed a number of additional major points raised in terms of literature supporting the important role of C/EBP α in modulating TGFB-1 – induced EMT, additional analyses / details, use of additional shRNA or siRNA Smartpools as an alternative validation of sh-RNA data, and most importantly, supporting evidence from clinical breast cancer metastasis data and breast cancer cell line systems. The Minor issues raised have also been addressed.

The manuscript is much improved / refined with the Responses to both reviewers, and I am happy with the corrections made to Reviewer 2. I have a number of minor points to make below:

Minor:

1. 2nd, 3rd, 4th authors contributed equally, but not 1st-? OK

This is correct.

2. Mention name of 'well established mouse model of emt-driven metastasis' in the abstract

The mouse model is now specified.

3. The transcription factors, known as EMT-effectors: There is a push for these to be called EMT-TFs; can the authors pre-empt this and adjust?

We have adjusted the text accordingly in the revised manuscript (page 3).

4. If C/EBP α levels are reduced in the majority of breast cancer specimens (17), why so little EMT in BrCa generally?

We propose that TGF- β -mediated EMT is only one of the possible mechanisms by which C/EBP α can be downregulated in breast cancer. For example, hypoxia has been shown to induce C/EBP α downregulation in breast cancer ¹. Additionally, the plastic nature of the EMT process can be sufficient to undermine the identification of the EMT programme in breast cancer patients. Therefore, we suggest that, and in line with breast cancer being such as heterogenous disease, there are multiple pro-oncogenic pathways that can downregulate C/EBP α and contribute to breast cancer progression, where TGF- β signaling plays a critical role.

5. Response to this optional. C/EBP α expression levels (mRNA) are not much reduced in human breast cancer cell lines in the Basal or Mesenchymal subgroups compared to Luminal subgroup (Daemon et al, 2003, PMID 24176112). This is, however, more change than seen in CEBP-beta, delta or gamma. It may be worth a comment. The modest (but apparent) reduction of CEBPA mRNA across these subgroups may support the proposed post-Transcriptional regulation mentioned in the Discussion. See also Figs S8c,d...

We thank the reviewer for raising this interesting point. This is In line with our results from **Figures S8c-d** where we observed a robust reduction in C/EBP α expression when we compare non-transformed breast cancer cell lines with transformed breast cancer cell lines. However, no significant changes are observed between MCF-7 cells (luminal) and MDA-MB-231 cells (basal-like). Contrasting cytoplasmic C/EBP α levels in these transformed cell lines can be one regulatory mechanism by which C/EBP α impacts differentially these subtypes, as well as perhaps different post-transcriptional events

might have a distinct impact on C/EBP α protein-protein interactions and transcriptional activity. Also, we cannot exclude that an EMT-like programme occurs in the luminal subtype of breast cancer since much of the work assessing the role of EMT in breast cancer has been centred around non-luminal subtypes. Although non-luminal breast cancers are typically characterized by more mesenchymal characteristics and more often associated with EMT, loss of the estrogen receptor function in luminal cancers has been shown to result in trans-differentiation from an epithelial to a mesenchymal phenotype². Additionally, neglectable SOX9 expression was described in numerous luminal breast cancer patients³. These observations can therefore explain the lack of a dramatic reduction of C/EBP α expression in luminal subtype compared to non-luminal subtypes as EMT-mediated downregulation of C/EBP α might occur in certain subtypes of luminal breast cancers.

6. In the newly added sentence “Cells were expanded for 2 weeks after sorting before experiments. The expression of C/EBP α p42 isoform was confirmed by Immunoblotting analysis.’ I don’t think Immunoblotting should be capitalized?

This has been changed in the revised manuscript.

7. shRNA targeting CEBPA (Sigma-Aldrich, TRCN000007304)... Thank-you for adding the siRNA smartpool date now. Later it states: “CEBPA-depleted HMLE cells were generated using two independent shRNA constructs” but that “expression showed strong reduction in CEBPA mRNA levels in only one of the shRNAs used”. The catalog number of the one that didn’t work so well should be included.

The information regarding the second shRNA against CEBPA has now been added to the M&M on page 19 in the revised manuscript.

8. Figure 5a; the statistical significance associated with * and ** is not defined? Please check all Figures for this.

Information regarding statistical significance is included in the “Statistical analysis” section on page 27 in the revised manuscript.

9. In the new text “we choose to further explore the role of C/EBP α as it has been shown to play a role in cell differentiation” I would suggest to add ‘especially’ at the start of the new text?

A sentence has been added in the revised manuscript (page 5).

10. Under “Wound healing migration assays” 24-well plates were coated with Collagen I, Collagen IV or Fibronectin” I think the proteins should not be capitalised?

This has been changed in the revised manuscript.

11. I think the following new statement “Mammary tumors from this model, and similar to human ductal carcinomas, consist of a very small percentage of E-cadherin low cells which makes detecting changes in E-cadherin low cell number challenging” could be references, e.g. ref 4? Fig 7B results should be described differently – If not significant it cannot be called lower? The most important result still holds - The reduced metastasis seen.

This reference has been added and wording has been modified in the revised manuscript (page 12).

12. In the Discussion, the following sentence “In accordance with these observations, constitutive expression of C/EBP α during TGF- β -mediated EMT contributed to repression of ZEB1 (Fig. S3b), N-cadherin and MMP2 as well as maintenance of E-cadherin levels, however no significant repression of vimentin was detected.” has a Figure reference for Zeb1 but not the other factors (Fig 4B?)

The figure reference has been added in the revised manuscript.

13. In the Discussion, the following statement is made: “Moreover, by using an established mice model of EMT-induced breast cancer, we validated the tumor suppressive role of C/EBP α in vivo.” However, Figure 7A clearly shows no effect on Primary tumor growth? Should this sentence read ‘metastasis’ rather than ‘tumor’? Metastasis is clearly reduced (Figure 7c, 7d)..

We have modified the wording in the revised manuscript.

14. Similarly, just further in the Discussion, it is stated ‘These data support the notion that C/EBP α is an inhibitory factor for tumor development and metastasis formation in a clinical setting. Should ‘tumor development’ be included here.

The wording has been modified in the revised manuscript.

15. Figure 8 indicates a cell exhibiting partial-EMT, with reduced C/EBP α ; what is the scenario anticipated in the full EMT shown in the adjacent cell?

C/EBP α protein levels are dramatically reduced to very low levels after 15 days of TGF- β (mimicking ‘full’ EMT). The remaining detectable C/EBP α is mostly found in the cytoplasm indicating that C/EBP α is excluded from the nucleus and therefore impeded of its transcriptional activity and consequently not able to exert its anti-metastatic functions.

References:

1. Seifeddine, R., Fulchignoni-Lataud, M.-C. & Massaad-Massade, L. Down-regulation of C/EBP α in breast cancer cells by hypoxia-estrogen combination is mainly due to hypoxia. *Anticancer Res.* **29**, 1227–31 (2009).

2. Al Saleh, S., Al Mulla, F. & Luqmani, Y. A. Estrogen Receptor Silencing Induces Epithelial to Mesenchymal Transition in Human Breast Cancer Cells. *PLoS One* **6**, e20610 (2011).
3. Pomp, V. *et al.* Differential expression of epithelial–mesenchymal transition and stem cell markers in intrinsic subtypes of breast cancer. *Breast Cancer Res. Treat.* **154**, 45–55 (2015).

REVIEWERS' COMMENTS:

Reviewer #1 (Remarks to the Author):

The authors have satisfactorily addressed my remaining concerns.

Reviewer #2 (Remarks to the Author):

The authors have sufficiently addressed the minor comments made at the last review. I appreciate their commentary in regards my optional point 5, and agree with their thinking. These are very interesting questions requiring future work.

Erik (Rik) Thompson